# Smoothing the Shift: Towards Stable Test-Time Adaptation under Complex Multimodal Noises

**Zirun Guo   Tao Jin**[*]
Zhejiang University
zrguo.cs@gmail.com

## Abstract

Test-Time Adaptation (TTA) aims to tackle distribution shifts using unlabeled test data without access to the source data. In the context of multimodal data, there are more complex noise patterns than unimodal data such as simultaneous corruptions for multiple modalities and missing modalities. Besides, in real-world applications, corruptions from different distribution shifts are always mixed. Existing TTA methods always fail in such multimodal scenario because the abrupt distribution shifts will destroy the prior knowledge from the source model, thus leading to performance degradation. To this end, we reveal a new challenge named *multimodal wild TTA*. To address this challenging problem, we propose two novel strategies: sample identification with interquartile range **S**moothing and **u**nimodal assistance, and **M**utual **i**nformation sharing (SuMi). SuMi smooths the adaptation process by interquartile range which avoids the abrupt distribution shifts. Then, SuMi fully utilizes the unimodal features to select low-entropy samples with rich multimodal information for optimization. Furthermore, mutual information sharing is introduced to align the information, reduce the discrepancies and enhance the information utilization across different modalities. Extensive experiments on two public datasets show the effectiveness and superiority over existing methods under the complex noise patterns in multimodal data. Code is available at https://github.com/zrguo/SuMi.

## 1 Introduction

Deep learning has achieved remarkable success and has been widely adopted across a variety of applications (Touvron et al., 2023; Podell et al., 2024; Yan et al., 2024; Lin et al., 2024). However, these models often struggle when faced with data distributions that differ from their training data. For example, in real-world scenarios, unexpected environmental changes and noises always occur such as weather changes and data corruption. When encountering such domain shifts, model performance can degrade rapidly (Hendrycks & Dietterich, 2019). To address this challenge, many adaptation techniques such as domain adaptation (Zhu et al., 2023) and domain generalization (Zhou et al., 2023a) have been proposed to enhance the robustness of models. One of the most challenging settings is Test-Time Adaptation (TTA) (Wang et al., 2021; Niu et al., 2022), where the model must adapt to a target domain without access to any source domain data and labels of target data. Recently, numerous promising test-time adaptation methods (Niu et al., 2023; Yang et al., 2024; Lee et al., 2024; Chen et al., 2024; Guo et al., 2024b) have shown great results.

However, the majority of existing TTA methods have focused on unimodal scenarios. In comparison to unimodal tasks, multimodal tasks often face more complex noise patterns, such as simultaneous noise corruption across multiple modalities or missing modalities. In this work, we broadly categorize multimodal noise scenarios into two types (shown in Figure 1(a)): weak Out-Of-Distribution (OOD) samples, where only one modality is corrupted by noise, and strong OOD samples, where multiple modalities are corrupted by noise or missing modality issues occur. Considering a multimodal sentiment analysis system, audio corruption can arise from factors like the speaker's accent,

---

[*]Corresponding author

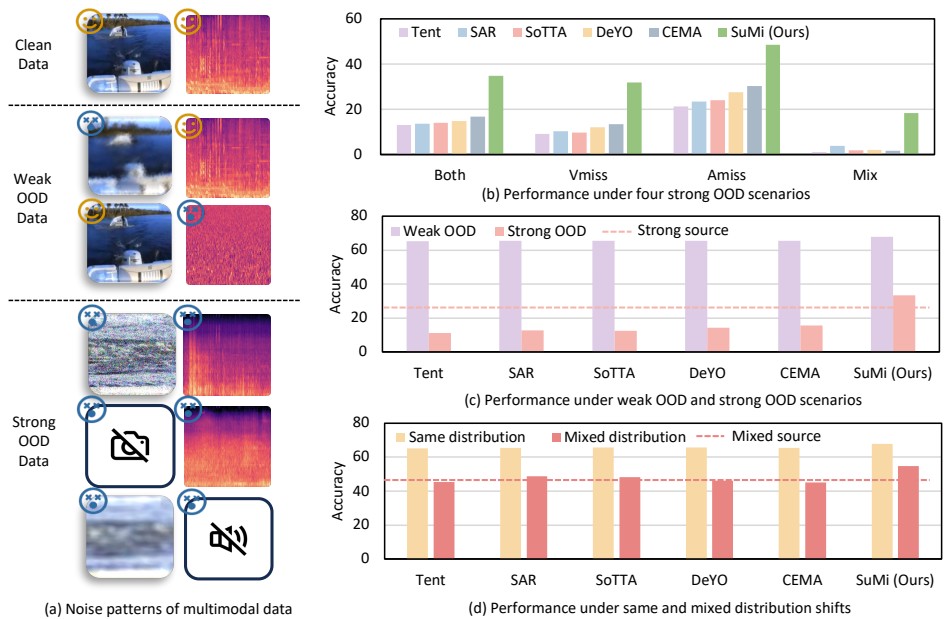

Figure 1: Illustration of our task where the target domain includes various domain shifts including weak OOD and strong OOD samples. The performances of existing methods degrade significantly on this challenging task, even worse than the source model. We get these results on Kinetics50-C.

dialect, and background noise. Video corruption can occur due to lighting variations and diverse facial features. Furthermore, audio corruption can lead to transcription errors in text, creating text noise and leading to simultaneous domain shifts (strong OOD) across modalities. As shown in Figure 1(b) and (c), we observe the performance of existing TTA methods can degrade significantly when faced with the more complex noise patterns encountered in multimodal scenarios, especially in the case of strong OOD samples. The huge distribution gap between the source domain and the strong OOD data will damage the prior knowledge in the source model, thus leading to performance degradation. Additionally, in real-world dynamic environments where the target domain includes various types of distribution shifts (known as wild TTA), the performance of existing TTA methods always fail (Niu et al., 2023). To address the challenge in wild TTA, Niu et al. (2023) propose a sharpness-aware and reliable entropy minimization method to further stabilize TTA. However, as shown in Figure 1(d), in the context of multimodal wild TTA where the target domain includes various distribution shifts, the results are still not satisfying. A recent work READ (Yang et al., 2024) explores the reliability bias in multimodal data during test time. READ proposes that when one of the modalities is corrupted, the reliability balance across the modalities will be destroyed, which leads to a heavy performance degradation of the model. However, it only discusses the weak OOD situations and overlooks the more complex noise patterns in multimodal data. Besides, it is based on the mild TTA setting where test samples have the same distribution shift type.

Based on the above observations and the limitations of existing methods, in this paper, we reveal a new challenging task named *multimodal wild TTA* where the target domain includes various types of distribution shifts including weak OOD samples and strong OOD samples. To address the challenging problem, we propose two novel strategies: sample identification with interquartile range **S**moothing and **u**nimodal assistance and **M**utual **i**nformation sharing (SuMi). To avoid the abrupt distribution shifts which could destroy the prior knowledge from the source model, we propose to smooth the adaptation process with interquartile range. Besides, we fully utilize the unimodal information to select low-entropy samples with rich multimodal information. Furthermore, we propose the mutual information sharing to align information between different modalities which can reduce the discrepancies across different modalities and enhance the information utilization of different modalities. Our main contributions can be summarized as follows:

- We show that the complex noise patterns in multimodal data will make existing TTA methods fail. To this end, we propose a new practical and challenging task named *multimodal*

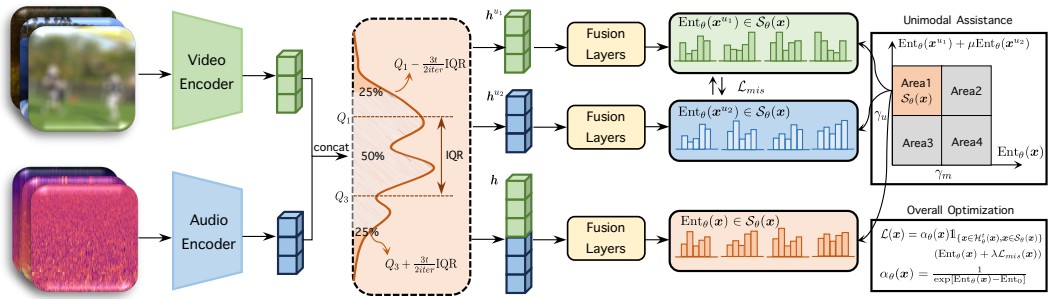

Figure 2: The overview of SuMi.

*wild TTA* where the target domain includes various types of distribution shifts including weak OOD samples and strong OOD samples.

- We propose a novel method SuMi, consisting of sample identification with interquartile range smoothing and unimodal assistance, and mutual information sharing.

- SuMi outperforms all the baselines consistently and significantly in weak, strong and mixed OOD domains. Additionally, we build two benchmarks for multimodal wild TTA.

## 2    RELATED WORK

Test-time adaptation aims to update the source model without source domain data and labels of the target domain data. Test-time training, such as TTT (Sun et al., 2020) and TTT+ (Liu et al., 2021) trains a source model with both supervised and self-supervised objectives in the training stage to enhance test-time adaptation. These methods depend on proxy tasks and assume that the training process is controllable, which limits the scope of applications. Therefore, fully test-time adaptation methods (Wang et al., 2021; Niu et al., 2022; Zhou et al., 2023b; Yuan et al., 2023; Gong et al., 2023a; Park et al., 2024) are proposed to adapt the model only in test-time, without intervening in the training stage. Tent (Wang et al., 2021) proposes to use entropy minimization to update the normalization layers of the model. Furthermore, EATA (Niu et al., 2022) and SAR (Niu et al., 2023) propose the sample selection criteria for entropy minimization. More recently, Lee et al. (2024) show that using entropy alone as a measure of confidence is insufficient and propose to use a combination of entropy and the proposed PLPD metric to identify samples. Chen et al. (2024) proposes a dynamic unreliable and low-informative sample exclusion method for entropy minimization.

However, existing works focus on the unimodal TTA. Compared to unimodal scenarios, multimodal data face much more complex patterns of noise in real-world applications, such as simultaneous corruptions and missing modalities (Guo et al., 2024c). Shin et al. (2022) proposes a framework to generate cross-modal pseudo labels as self-training signals. Guo et al. (2024b) propose a multimodal TTA approach but focus on the multimodal regression tasks. A recent work (Yang et al., 2024) explores the multimodal TTA and proposes reliable fusion and robust adaptation to address information discrepancies in multimodal data. However, it only discusses the situations where there is only one modality corrupted. When there are multiple modalities corrupted or missing, the huge and abrupt distribution gap between the source domain and the target domain will make the method fail. Additionally, it focuses on the single domain adaptation. In contrast, we explore a more practical and challenging wild TTA where the target domain includes various types of corruption.

## 3    METHODOLOGY

### 3.1    PROBLEM FORMULATION

Without loss of generality, we take two modalities as an example for clarity of presentation. Let $\mathcal{M}_\theta = (\phi_{u_1}, \phi_{u_2}, \mathcal{F})$ with parameter $\theta$ be the source model pre-trained on the source domain dataset $\mathcal{D}_{source} = \{(\boldsymbol{x}_i, y_i)\}_{i=1}^{N_s}$ where $\phi_{u_1}, \phi_{u_2}$ are the encoders of modality $u_1$ and $u_2$, $\mathcal{F}$ is the multi-

modal fusion layers with prediction head, $\boldsymbol{x}_i = (\boldsymbol{x}_i^{u_1}, \boldsymbol{x}_i^{u_2})$, and $N_s$ is the number of samples. TTA aims to fine-tune the source model $\mathcal{M}_\theta$ on the target domain dataset $\mathcal{D}_{target} = \{\boldsymbol{x}_i\}_{i=1}^{N_t}$ where the labels and the source dataset are unavailable. Existing TTA methods (Niu et al., 2023; Yang et al., 2024) update the parameter $\theta$ by minimizing the entropy of test domain data:

$$\mathrm{Ent}_\theta(\boldsymbol{x}) = -\mathbf{p}_\theta(\boldsymbol{x}) \log \mathbf{p}_\theta(\boldsymbol{x}) = -\sum_{i=1}^{C} p_\theta(\boldsymbol{x})_i \log p_\theta(\boldsymbol{x})_i \tag{1}$$

where $\mathbf{p}_\theta = \mathrm{softmax}(\mathcal{M}_\theta(\boldsymbol{x})) = (p_\theta(\boldsymbol{x})_1, p_\theta(\boldsymbol{x})_2, \cdots, p_\theta(\boldsymbol{x})_C)$ is the probabilistic distribution outputted by the model $\mathcal{M}_\theta$ and $C$ is the number of classes.

In this paper, we reveal a new challenging task named *multimodal wild TTA*. Specifically, we broadly categorize multimodal noise scenarios into two types: weak OOD samples, where only one modality is corrupted by noise, and strong OOD samples, where multiple modalities are corrupted by noise or missing modality issues occur. Multimodal wild TTA considers a more practical and challenging TTA setting where the target datasets contain various types of distribution shifts including both weak OOD samples and strong OOD samples. The overall architecture of SuMi is presented in Figure 2.

### 3.2 SAMPLE IDENTIFICATION WITH INTERQUARTILE RANGE SMOOTHING AND UNIMODAL ASSISTANCE

#### 3.2.1 INTERQUARTILE RANGE SMOOTHING

Many existing TTA methods rely on selecting low-entropy samples for entropy minimization (Niu et al., 2022; Lee et al., 2024; Chen et al., 2024). However, when the target data is a mixture of various types of distribution shifts, including weak OOD samples and strong OOD samples, the performance of the model would degrade significantly. For example, in Figure 3(a), we present the performance of three different settings of the adaptation process. We can observe that directly adapting the model to the strong OOD domain will yield much poorer performance than a model adapted on weak OOD domain. The main reason is that there is a huge distribution gap between the source domain and the strong OOD domain. Therefore, a direct adaptation would destroy the prior knowledge of the source model and lead to instability. In comparison, when we first perform adaptation on the weak OOD domain before the strong OOD domain, the performance of the model will improve. This phenomenon inspires us that a smoothing adaptation process under the wild TTA and complex noise patterns of multimodal data is much better than an abrupt adaptation process.

Motivated by the above observations, we propose an interquartile range smoothing method for dynamic sample identification during the adaptation process. Interquartile range (IQR) is a measure of statistical dispersion, which is the spread of the data (Dekking et al., 2006). We give a brief definition of IQR below:

**Definition 1** *IQR is the difference between the 75th and 25th percentiles of the data. The data is divided into four rank-ordered even parts via linear interpolation which are denoted as $Q_1$ (lower quartile), $Q_2$ (median) and $Q_3$ (upper quartile). IQR is calculated as $IQR = Q_3 - Q_1$.*

IQR is often used to identify unstable samples or outliers in a dataset. Specifically, according to Tukey's rule (Tukey et al., 1977), the stable sample set $\mathcal{X}_s$ is selected as:

$$\mathcal{X}_s = \{x \mid x \geq Q_1 - \frac{3}{2}\mathrm{IQR} \text{ and } x \leq Q_3 + \frac{3}{2}\mathrm{IQR}\} \tag{2}$$

To smooth the adaptation process, we modify the above equation slightly and select the samples $\mathcal{H}_\theta^t(\boldsymbol{x})$ as:

$$\mathcal{H}_\theta^t(\boldsymbol{x}) = \{\boldsymbol{h} \mid \boldsymbol{h} \geq Q_1 - \frac{3}{2}f(t)\mathrm{IQR} \text{ and } \boldsymbol{h} \leq Q_3 + \frac{3}{2}f(t)\mathrm{IQR}\}$$
$$\boldsymbol{h} = [\boldsymbol{h}^{u_1}, \boldsymbol{h}^{u_2}], \boldsymbol{h}^{u_1} = \phi_{u_1}(\boldsymbol{x}^{u_1}), \boldsymbol{h}^{u_2} = \phi_{u_2}(\boldsymbol{x}^{u_2}) \tag{3}$$

where $t$ is the current iteration, $\theta$ is the parameter of the model, $f(t)$ is the smoothing function, $[,]$ is the concatenation operation and $\boldsymbol{h}$ is the representation of the sample. For simplicity, we use the linear smoothing and set $f(t) = t/iter$ where $iter$ is the total iteration. We use the representations instead of the raw inputs because the representations are informative dense vectors that contain less

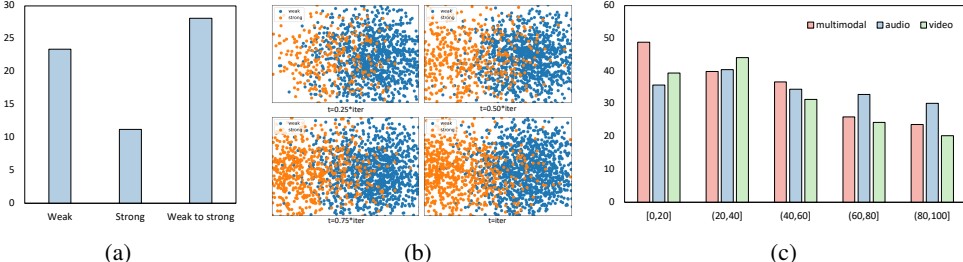

Figure 3: (a) Performance of different adaptation settings on strong OOD samples. (b) t-SNE visualizations (Van der Maaten & Hinton, 2008) of features during adaptation. (c) Performance using different quantiles multimodal and unimodal entropy. Results are obtained on Kinetics50-C.

noise and unrelated information than the raw inputs. At iteration $t$, we use the selected data $\mathcal{H}_\theta^t(\boldsymbol{x})$ for adaptation. In Figure 3(b), we visualize the sample identification process using the source model. From the figure, we can observe that at first several iterations, most weak OOD samples are selected. With the increase of $t$, the data for adaptation is also increasing, including more and more strong OOD samples. This smoothing process enables gradual adaptation to the strong OOD samples and various types of distribution shifts, avoiding the abrupt distribution gaps which could destroy the prior knowledge of the source model. Additionally, $\boldsymbol{h}$ is a vector. Therefore, in practice, we select $\boldsymbol{h}$ for adaptation if $\beta + (1 - \beta)f(t)$ percent of the values in $\boldsymbol{h}$ satisfy Equation 3 for stability.

### 3.2.2 UNIMODAL ASSISTANCE

IQR smoothing aims to help the model preserve the prior knowledge in the source model and gradually adapt to the strong OOD samples and various types of distribution shifts. However, this process can not distinguish the high-quality samples that benefit the entropy minimization. Therefore, we introduce a novel sample identification method for multimodal data. As suggested in previous work (Niu et al., 2022; 2023; Chen et al., 2024), low-entropy samples will benefit the entropy minimization while high-entropy samples, due its uncertainty, will adversely affect the process. However, in the context of multimodal data, apart from multimodal entropy, there is unimodal entropy we can utilize to help the adaptation. As shown in Figure 3(c), we conduct experiments using unimodal entropy and multimodal entropy. We can easily observe that the multimodal low-entropy samples will yield much better performance than high-entropy samples. However, for audio and video modality, the samples of $(20, 40]$ interval yield better results than samples of $[0, 20]$. This indicates that for unimodal entropy, lower entropy does not necessarily correlate with better performance. When only one modality exhibits low entropy, it indicates that the sample does not rely on multimodal data for accurate prediction, implying it has low informative value for multimodal optimization. Conversely, a unimodal sample with slightly higher entropy shows a dependence on multimodal data for accurate prediction, indicating that it contains rich multimodal information.

Inspired by the above observations, we propose a sample identification method with unimodal assistance to select low-entropy samples with rich multimodal information. Specifically, our method employs the following identification criteria:

$$\mathcal{S}_\theta(\boldsymbol{x}) = \{\boldsymbol{x} \mid \text{Ent}_\theta(\boldsymbol{x}) \leq \gamma_m \text{ and } (\text{Ent}_\theta(\boldsymbol{x}^{u_1}) + \mu\text{Ent}_\theta(\boldsymbol{x}^{u_2})) \geq \gamma_u\} \tag{4}$$

where $\gamma_m$ and $\gamma_u$ are the pre-defined threshold for multimodal and unimodal entropy and $\mu$ is a trade-off between modalities. By limiting the multimodal entropy, we can select low-entropy samples with high certainty and fewer noises. Meanwhile, by limiting the unimodal entropy, we can ensure the samples selected contain rich multimodal information, excluding low-informative samples.

### 3.3 MUTUAL INFORMATION SHARING

A recent study (Yang et al., 2024) reveals a challenge in multimodal TTA, known as reliability bias, which refers to the information discrepancies across different modalities. During the adaptation process, this discrepancy often leads to imbalanced utilization of each modality (Guo et al., 2024a). In strong OOD situations, missing modality cases could occur or multiple modalities could be corrupted. Therefore, the imbalanced phenomenon could be enlarged. How to balance the adaptation

---

**Algorithm 1** SuMi

---

1: **Input:** Source model $\mathcal{M}_\theta = (\phi_{u_1}, \phi_{u_2}, \mathcal{F})$, target dataset $\mathcal{D}_{target} = \{\boldsymbol{x}_i\}_{i=1}^{N_t}$, adaptation iterations $T$ and a series of hyperparameters.
2: **for** $t = 1$ **to** $T$ **do**
3:     $\boldsymbol{x} = (\boldsymbol{x}^{u_1}, \boldsymbol{x}^{u_2}) \overset{\text{Sample}}{\longleftarrow} \mathcal{D}_{target}$;
4:     Calculate the representations $\boldsymbol{h} = [\boldsymbol{h}^{u_1}, \boldsymbol{h}^{u_2}], \boldsymbol{h}^{u_1} = \phi_{u_1}(\boldsymbol{x}^{u_1}), \boldsymbol{h}^{u_2} = \phi_{u_2}(\boldsymbol{x}^{u_2})$;
5:     Calculate $Q_1$ and $Q_3$: $Q_1 = \text{quantile}(\boldsymbol{h}, 0.25), Q_3 = \text{quantile}(\boldsymbol{h}, 0.75)$;
6:     Calculate IQR: $\text{IQR} = Q_3 - Q_1$;
7:     Select samples $\mathcal{H}_\theta^t(\boldsymbol{x})$ using Equation 3;
8:     Calculate entropy of multimodal outputs and unimodal outputs;
9:     Select samples $\mathcal{S}_\theta(\boldsymbol{x})$ from $\mathcal{H}_\theta^t(\boldsymbol{x})$ using Equation 4;
10:    Calculate the entropy in $\mathcal{S}_\theta(\boldsymbol{x})$;
11:    **if** $t < t_0$ **then**
12:       Calculate mutual information sharing loss using Equation 6;
13:    **end if**
14:    Calculate the loss $\mathcal{L}(\boldsymbol{x})$ using Equation 8;
15:    Update the affine parameters of the model $\mathcal{M}_\theta$;
16: **end for**

---

across different modalities under strong OOD domains (especially missing modality cases) is very important. To address this problem, we propose a simple yet effective method, mutual information sharing, to align information between different modalities. Concretely, for modality $u_i$, we can obtain its probabilistic distribution as $\mathbf{p}_\theta^{u_i}(\boldsymbol{x}^{u_i}) = \text{softmax}(\mathcal{F}(\phi_{u_i}(\boldsymbol{x}^{u_i})))$. For simplicity, we will use $\mathbf{p}^{u_i}$ to represent $\mathbf{p}_\theta^{u_i}(\boldsymbol{x}^{u_i})$ in the following context. We define the complementary probabilistic distribution of $\mathbf{p}^{u_i}$ as

$$\mathbf{p}^{u_i\prime} = (\sum_{j=1}^{M} \mathbf{p}^{u_j} - \mathbf{p}^{u_i})/(M-1) \tag{5}$$

where $M$ is the number of modalities. For two modalities, $\mathbf{p}^{u_1\prime} = \mathbf{p}^{u_2}$ and $\mathbf{p}^{u_2\prime} = \mathbf{p}^{u_1}$. To improve the alignment between different modalities, we can minimize the *KL divergence* (Kullback & Leibler, 1951) between the probabilistic distribution $\mathbf{p}^{u_i}$ and its complementary distribution $\mathbf{p}^{u_i\prime}$. However, if one modality is severely corrupted, minimizing the KL divergence might influence the clean modality. Therefore, we add multimodal distribution $\mathbf{p}^m = \text{softmax}(\mathcal{M}_\theta(\boldsymbol{x}))$ to improve the robustness and stability. Therefore, we can represent the mutual information sharing loss as:

$$
\begin{aligned}
\mathcal{L}_{mis}(\boldsymbol{x}) &= D_{KL}(\mathbf{p}^{u_1} \parallel \frac{1}{2}(\mathbf{p}^{u_1\prime} + \mathbf{p}^m)) + D_{KL}(\mathbf{p}^{u_2} \parallel \frac{1}{2}(\mathbf{p}^{u_2\prime} + \mathbf{p}^m)) \\
&= \sum_{i=1}^{C} p_i^{u_1} \log \frac{2p_i^{u_1}}{p_i^{u_1\prime} + p_i^m} + \sum_{i=1}^{C} p_i^{u_2} \log \frac{2p_i^{u_2}}{p_i^{u_2\prime} + p_i^m}
\end{aligned} \tag{6}
$$

where $C$ is the number of classes and $p_i$ is the $i$-th value of $\mathbf{p}$. Mutual information sharing can help the model connect and align the information between different modalities. Through mutual information sharing, when there are corrupted modalities including missing modalities, information from other modalities could be utilized to enhance the predictions.

## 3.4 OVERALL OPTIMIZATION

Following previous TTA methods (Niu et al., 2022; Lee et al., 2024), we add a weighting term to emphasize the contributions of samples during adaptation. Specifically, the weighting term is calculated as

$$\alpha_\theta(\boldsymbol{x}) = \frac{1}{\exp[\text{Ent}_\theta(\boldsymbol{x}) - \text{Ent}_0]} \tag{7}$$

where $\text{Ent}_0$ is a pre-defined normalization factor (Niu et al., 2022). In summary, we can denote the overall loss function as:

$$\mathcal{L}(\boldsymbol{x}) = \alpha_\theta(\boldsymbol{x}) \mathbb{1}_{\{\boldsymbol{x} \in \mathcal{H}_\theta^t(\boldsymbol{x}), \boldsymbol{x} \in \mathcal{S}_\theta(\boldsymbol{x})\}} (\text{Ent}_\theta(\boldsymbol{x}) + \lambda \mathcal{L}_{mis}(\boldsymbol{x})) \tag{8}$$

Table 1: Accuracy comparison with SOTA methods on Kinetics50-C with corrupted video modality (severity level 5). We report $\text{avg}_{\pm\text{std}}$ over five random seeds. **Bold**: best results. Underline: second best results.

| | Noise | | | Blur | | | | Weather | | | | Digital | | | | Avg. |
|---|---|---|---|---|---|---|---|---|---|---|---|---|---|---|---|---|
| | Gauss. | Shot | Impul. | Defoc. | Glass | Motion | Zoom | Snow | Frost | Fog | Brit. | Contr. | Elastic | Pixel | JPEG | |
| Source | 47.7 | 48.8 | 47.4 | 67.4 | 61.0 | 71.1 | 66.1 | 60.7 | 62.1 | 45.5 | 75.9 | 51.9 | 65.1 | 67.8 | 63.7 | 60.2 |
| • Tent | $48.3_{\pm0.4}$ | $49.3_{\pm0.7}$ | $48.4_{\pm0.5}$ | $67.8_{\pm0.3}$ | $62.2_{\pm0.5}$ | $\underline{71.9}_{\pm0.3}$ | $67.8_{\pm0.5}$ | $63.1_{\pm0.5}$ | $63.5_{\pm0.5}$ | $23.0_{\pm1.0}$ | $75.9_{\pm0.4}$ | $50.1_{\pm0.2}$ | $67.8_{\pm0.3}$ | $70.5_{\pm0.1}$ | $67.1_{\pm0.3}$ | 59.8 |
| • EATA | $49.0_{\pm0.2}$ | $50.3_{\pm0.3}$ | $49.2_{\pm0.1}$ | $67.9_{\pm0.1}$ | $63.9_{\pm0.4}$ | $71.6_{\pm0.3}$ | $67.9_{\pm0.3}$ | $63.4_{\pm0.2}$ | $64.4_{\pm0.1}$ | $47.0_{\pm0.3}$ | $76.1_{\pm0.1}$ | $52.2_{\pm0.3}$ | $67.6_{\pm0.2}$ | $70.3_{\pm0.2}$ | $\underline{67.9}_{\pm0.2}$ | 61.9 |
| • SAR | $48.6_{\pm0.3}$ | $49.8_{\pm0.6}$ | $48.7_{\pm0.6}$ | $67.9_{\pm0.2}$ | $62.7_{\pm0.4}$ | $71.8_{\pm0.4}$ | $67.9_{\pm0.5}$ | $63.4_{\pm0.4}$ | $64.2_{\pm0.3}$ | $24.0_{\pm0.9}$ | $75.9_{\pm0.4}$ | $51.1_{\pm0.3}$ | $68.0_{\pm0.1}$ | $70.6_{\pm0.2}$ | $67.2_{\pm0.2}$ | 60.1 |
| • SoTTA | $48.3_{\pm0.3}$ | $49.8_{\pm0.2}$ | $48.5_{\pm0.4}$ | $67.9_{\pm0.2}$ | $62.5_{\pm0.3}$ | $\underline{71.9}_{\pm0.3}$ | $67.8_{\pm0.6}$ | $63.2_{\pm0.5}$ | $64.0_{\pm0.2}$ | $27.6_{\pm1.1}$ | $75.7_{\pm0.3}$ | $51.3_{\pm0.4}$ | $67.8_{\pm0.3}$ | $70.4_{\pm0.2}$ | $67.5_{\pm0.4}$ | 60.3 |
| • DeYO | $48.6_{\pm0.3}$ | $49.8_{\pm0.6}$ | $48.7_{\pm0.6}$ | $\underline{68.0}_{\pm0.3}$ | $63.0_{\pm0.4}$ | $\underline{71.9}_{\pm0.3}$ | $68.1_{\pm0.5}$ | $63.5_{\pm0.4}$ | $64.4_{\pm0.4}$ | $21.4_{\pm1.0}$ | $75.9_{\pm0.4}$ | $50.6_{\pm0.2}$ | $68.6_{\pm0.4}$ | $\underline{70.8}_{\pm0.2}$ | $67.5_{\pm0.3}$ | 60.0 |
| • CEMA | $48.4_{\pm0.3}$ | $49.4_{\pm0.5}$ | $48.6_{\pm0.6}$ | $67.8_{\pm0.2}$ | $62.7_{\pm0.4}$ | $71.7_{\pm0.3}$ | $67.8_{\pm0.5}$ | $63.4_{\pm0.4}$ | $64.2_{\pm0.3}$ | $22.8_{\pm0.8}$ | $75.7_{\pm0.4}$ | $50.7_{\pm0.3}$ | $67.9_{\pm0.5}$ | $70.5_{\pm0.2}$ | $67.3_{\pm0.3}$ | 59.9 |
| • READ | $\underline{49.9}_{\pm0.5}$ | $\mathbf{50.8}_{\pm0.5}$ | $\underline{49.8}_{\pm0.7}$ | $67.9_{\pm0.5}$ | $\underline{65.1}_{\pm0.2}$ | $\mathbf{72.2}_{\pm0.2}$ | $\underline{69.2}_{\pm0.6}$ | $\underline{64.8}_{\pm0.5}$ | $\underline{66.7}_{\pm0.3}$ | $\underline{56.8}_{\pm0.2}$ | $\underline{76.2}_{\pm0.3}$ | $\underline{54.8}_{\pm0.4}$ | $\underline{68.9}_{\pm0.5}$ | $70.7_{\pm0.2}$ | $\mathbf{68.9}_{\pm0.2}$ | 63.5 |
| • SuMi | $\mathbf{50.1}_{\pm0.4}$ | $\underline{50.7}_{\pm0.3}$ | $\mathbf{50.4}_{\pm0.3}$ | $\mathbf{68.2}_{\pm0.3}$ | $\mathbf{65.6}_{\pm0.3}$ | $\mathbf{72.2}_{\pm0.2}$ | $\mathbf{69.7}_{\pm0.4}$ | $\mathbf{65.7}_{\pm0.3}$ | $\mathbf{67.0}_{\pm0.2}$ | $\mathbf{56.5}_{\pm0.5}$ | $\mathbf{77.1}_{\pm0.2}$ | $\mathbf{55.2}_{\pm0.4}$ | $\mathbf{69.3}_{\pm0.2}$ | $\mathbf{71.2}_{\pm0.2}$ | $\mathbf{68.9}_{\pm0.2}$ | **63.9** |

Table 2: Accuracy comparison with SOTA methods on Kinetics50-C with corrupted audio modality and strong OOD scenarios (severity level 5).

| | Noise | | | Weather | | | | Strong OOD | | | | |
|---|---|---|---|---|---|---|---|---|---|---|---|---|
| | Gauss. | Traff. | Crowd | Rain | Thund. | Wind | Avg. | Both | Vmiss | Amiss | Mix | Avg. |
| Source | 74.9 | 65.4 | 67.9 | 70.0 | 68.5 | 70.7 | 69.6 | 30.8 | 27.9 | 44.5 | 16.9 | 30.0 |
| • Tent | $74.8_{\pm0.5}$ | $68.2_{\pm0.5}$ | $\underline{70.3}_{\pm0.3}$ | $71.1_{\pm0.4}$ | $66.7_{\pm0.5}$ | $71.7_{\pm0.1}$ | 70.5 | $13.1_{\pm0.4}$ | $9.2_{\pm0.5}$ | $21.3_{\pm0.4}$ | $1.2_{\pm0.5}$ | 11.2 |
| • EATA | $\underline{74.9}_{\pm0.1}$ | $68.0_{\pm0.2}$ | $70.0_{\pm0.2}$ | $71.2_{\pm0.3}$ | $70.0_{\pm0.3}$ | $71.3_{\pm0.1}$ | 70.9 | $\underline{32.3}_{\pm0.2}$ | $\underline{28.6}_{\pm0.3}$ | $\underline{45.3}_{\pm0.2}$ | $14.9_{\pm0.3}$ | 30.3 |
| • SAR | $74.8_{\pm0.5}$ | $68.4_{\pm0.4}$ | $\underline{70.3}_{\pm0.3}$ | $71.2_{\pm0.4}$ | $68.9_{\pm0.4}$ | $71.9_{\pm0.1}$ | 70.9 | $13.6_{\pm0.4}$ | $10.3_{\pm0.4}$ | $23.5_{\pm0.4}$ | $3.8_{\pm0.5}$ | 12.8 |
| • SoTTA | $74.8_{\pm0.5}$ | $68.4_{\pm0.4}$ | $70.1_{\pm0.3}$ | $71.1_{\pm0.3}$ | $69.2_{\pm0.4}$ | $71.8_{\pm0.2}$ | 70.9 | $14.1_{\pm0.4}$ | $9.8_{\pm0.5}$ | $24.1_{\pm0.3}$ | $1.9_{\pm0.5}$ | 12.5 |
| • DeYO | $74.8_{\pm0.5}$ | $68.6_{\pm0.4}$ | $\underline{70.3}_{\pm0.4}$ | $71.3_{\pm0.5}$ | $70.4_{\pm0.3}$ | $\underline{72.0}_{\pm0.1}$ | 71.2 | $14.9_{\pm0.4}$ | $12.1_{\pm0.3}$ | $27.6_{\pm0.4}$ | $2.1_{\pm0.5}$ | 14.2 |
| • CEMA | $74.8_{\pm0.4}$ | $67.8_{\pm0.4}$ | $69.5_{\pm0.4}$ | $71.1_{\pm0.4}$ | $70.5_{\pm0.3}$ | $71.6_{\pm0.3}$ | 70.9 | $16.9_{\pm0.4}$ | $13.4_{\pm0.4}$ | $30.3_{\pm0.3}$ | $1.8_{\pm0.6}$ | 15.6 |
| • READ | $\underline{74.9}_{\pm0.5}$ | $\mathbf{69.1}_{\pm0.4}$ | $\underline{70.3}_{\pm0.2}$ | $\underline{71.4}_{\pm0.4}$ | $\mathbf{72.8}_{\pm0.5}$ | $71.3_{\pm0.3}$ | 71.6 | $31.1_{\pm0.3}$ | $27.5_{\pm0.5}$ | $44.3_{\pm0.3}$ | $13.7_{\pm0.3}$ | 29.1 |
| • SuMi | $\mathbf{75.1}_{\pm0.3}$ | $\underline{68.9}_{\pm0.3}$ | $\mathbf{70.6}_{\pm0.3}$ | $\mathbf{71.6}_{\pm0.3}$ | $\mathbf{72.8}_{\pm0.4}$ | $\mathbf{72.1}_{\pm0.2}$ | **71.9** | $\mathbf{34.8}_{\pm0.3}$ | $\mathbf{31.8}_{\pm0.3}$ | $\mathbf{48.6}_{\pm0.2}$ | $\mathbf{18.4}_{\pm0.4}$ | **33.4** |

Table 3: Accuracy comparison with SOTA methods on VGGSound-C with corrupted video modality (severity level 5).

| | Noise | | | Blur | | | | Weather | | | | Digital | | | | Avg. |
|---|---|---|---|---|---|---|---|---|---|---|---|---|---|---|---|---|
| | Gauss. | Shot | Impul. | Defoc. | Glass | Motion | Zoom | Snow | Frost | Fog | Brit. | Contr. | Elastic | Pixel | JPEG | |
| Source | 53.0 | 52.9 | 53.0 | 57.2 | 57.3 | 58.6 | 57.5 | 56.3 | 56.4 | 55.4 | 59.2 | 53.7 | 57.2 | 56.4 | 57.3 | 56.1 |
| • Tent | $53.0_{\pm0.1}$ | $53.2_{\pm0.1}$ | $52.9_{\pm0.1}$ | $56.3_{\pm0.1}$ | $56.3_{\pm0.1}$ | $57.6_{\pm0.1}$ | $56.8_{\pm0.1}$ | $55.4_{\pm0.1}$ | $56.0_{\pm0.1}$ | $56.2_{\pm0.1}$ | $58.3_{\pm0.1}$ | $53.5_{\pm0.1}$ | $57.3_{\pm0.1}$ | $56.7_{\pm0.1}$ | $56.8_{\pm0.1}$ | 55.8 |
| • EATA | $\underline{53.5}_{\pm0.1}$ | $\underline{53.7}_{\pm0.1}$ | $\underline{53.5}_{\pm0.1}$ | $57.1_{\pm0.1}$ | $57.1_{\pm0.0}$ | $58.2_{\pm0.1}$ | $57.7_{\pm0.1}$ | $56.0_{\pm0.1}$ | $56.6_{\pm0.1}$ | $56.7_{\pm0.1}$ | $\mathbf{59.4}_{\pm0.1}$ | $54.3_{\pm0.1}$ | $\underline{58.1}_{\pm0.2}$ | $\underline{57.3}_{\pm0.0}$ | $\underline{57.5}_{\pm0.1}$ | 56.5 |
| • SAR | $52.9_{\pm0.1}$ | $53.1_{\pm0.1}$ | $52.9_{\pm0.1}$ | $56.3_{\pm0.1}$ | $56.2_{\pm0.2}$ | $57.4_{\pm0.1}$ | $56.7_{\pm0.1}$ | $55.3_{\pm0.1}$ | $56.0_{\pm0.1}$ | $56.2_{\pm0.1}$ | $58.2_{\pm0.1}$ | $53.5_{\pm0.1}$ | $57.4_{\pm0.1}$ | $56.7_{\pm0.1}$ | $56.8_{\pm0.1}$ | 55.7 |
| • SoTTA | $52.9_{\pm0.1}$ | $53.2_{\pm0.1}$ | $52.9_{\pm0.1}$ | $56.6_{\pm0.1}$ | $56.8_{\pm0.2}$ | $57.9_{\pm0.2}$ | $57.1_{\pm0.1}$ | $55.7_{\pm0.1}$ | $56.1_{\pm0.1}$ | $56.3_{\pm0.1}$ | $\mathbf{59.4}_{\pm0.1}$ | $53.8_{\pm0.2}$ | $57.6_{\pm0.1}$ | $56.2_{\pm0.1}$ | $56.7_{\pm0.1}$ | 55.9 |
| • DeYO | $53.0_{\pm0.1}$ | $53.1_{\pm0.1}$ | $53.0_{\pm0.1}$ | $56.5_{\pm0.1}$ | $56.5_{\pm0.1}$ | $57.7_{\pm0.1}$ | $56.9_{\pm0.1}$ | $55.4_{\pm0.1}$ | $56.0_{\pm0.1}$ | $56.3_{\pm0.2}$ | $58.5_{\pm0.1}$ | $53.6_{\pm0.1}$ | $57.6_{\pm0.1}$ | $57.0_{\pm0.0}$ | $57.0_{\pm0.1}$ | 55.9 |
| • CEMA | $52.8_{\pm0.1}$ | $52.9_{\pm0.2}$ | $52.9_{\pm0.1}$ | $56.5_{\pm0.1}$ | $56.4_{\pm0.1}$ | $57.6_{\pm0.1}$ | $56.8_{\pm0.1}$ | $55.4_{\pm0.1}$ | $56.2_{\pm0.1}$ | $56.2_{\pm0.1}$ | $58.4_{\pm0.1}$ | $53.5_{\pm0.1}$ | $57.8_{\pm0.1}$ | $56.8_{\pm0.0}$ | $56.9_{\pm0.1}$ | 55.8 |
| • READ | $52.9_{\pm0.1}$ | $52.8_{\pm0.2}$ | $52.8_{\pm0.1}$ | $\underline{57.2}_{\pm0.2}$ | $\underline{57.3}_{\pm0.2}$ | $\underline{58.8}_{\pm0.2}$ | $\underline{58.1}_{\pm0.2}$ | $\underline{56.4}_{\pm0.1}$ | $\underline{57.5}_{\pm0.2}$ | $\underline{57.4}_{\pm0.1}$ | $\underline{59.3}_{\pm0.1}$ | $\underline{54.4}_{\pm0.2}$ | $57.8_{\pm0.1}$ | $\underline{56.6}_{\pm0.1}$ | $57.2_{\pm0.2}$ | 56.4 |
| • SuMi | $\mathbf{54.0}_{\pm0.1}$ | $\mathbf{54.3}_{\pm0.1}$ | $\mathbf{53.8}_{\pm0.1}$ | $\mathbf{58.2}_{\pm0.2}$ | $\mathbf{58.4}_{\pm0.1}$ | $\mathbf{59.4}_{\pm0.2}$ | $\mathbf{58.7}_{\pm0.1}$ | $\mathbf{57.5}_{\pm0.1}$ | $\mathbf{58.2}_{\pm0.1}$ | $\mathbf{57.6}_{\pm0.1}$ | $\mathbf{59.4}_{\pm0.1}$ | $\mathbf{54.8}_{\pm0.1}$ | $\mathbf{59.0}_{\pm0.1}$ | $\mathbf{57.5}_{\pm0.1}$ | $\mathbf{58.2}_{\pm0.1}$ | **57.3** |

where $\mathbb{1}_{\{\cdot\}}(\cdot)$ is an indicator function and $\lambda$ is a trade-off between the two losses. One point worth emphasizing is that for strong OOD adaptation, we only add the mutual information sharing loss $\mathcal{L}_{mis}$ in the first $t_0$ iterations during the adaptation process. The reason is that with the increase of iteration, the IQR smoothing will include more and more strong OOD samples where multiple modalities are corrupted which could damage the information sharing and the performance of the model. For weak OOD adaptation, we add mutual information sharing loss for all the iterations. Overall, Algorithm 1 presents the outline of our method.

## 4 EXPERIMENTS

### 4.1 EXPERIMENTAL SETTINGS

**Datasets.** We use two widely used multimodal datasets, Kinetics50 (Kay et al., 2017) and VG-GSound (Chen et al., 2020) for evaluation. Following previous work (Hendrycks & Dietterich, 2019; Yang et al., 2024), we introduce 15 different types of corruptions and 6 types for audio to simulate the distribution shifts in real-world applications. Each type of corruption has five levels of severity. For strong OOD samples, we introduce four different types: Both (both modalities are corrupted), Vmiss (video modality is missing), Amiss (audio modality is missing), and Mix (one modality is missing and the other is corrupted). Details of datasets and the corruptions are presented in Appendix A. As a result, we can obtain the corrupted datasets Kinetics50-C and VGGSound-C.

**Implementation Details.** For the source model, we use the pre-trained CAV-MAE (Gong et al., 2023b) following Yang et al. (2024). We use Adam optimizer with a learning rate of 1e-4/1e-5 and

Table 4: Accuracy comparison with SOTA methods on VGGsound-C with corrupted audio modality and strong OOD scenarios (severity level 5). We report avg$_{\pm\text{std}}$ over five random seeds. **Bold**: best results. Underline: second best results.

| | Noise | | | Weather | | | | Strong OOD | | | | |
| --- | --- | --- | --- | --- | --- | --- | --- | --- | --- | --- | --- | --- |
| | Gauss. | Traff. | Crowd | Rain | Thund. | Wind | Avg. | Both | Vmiss | Amiss | Mix | Avg. |
| Source | 37.2 | 21.2 | 16.9 | 21.8 | 27.4 | 25.6 | 25.0 | 9.4 | 28.0 | 18.9 | 6.0 | 15.6 |
| ● Tent | $6.0_{\pm0.3}$ | $1.6_{\pm0.1}$ | $1.1_{\pm0.0}$ | $1.7_{\pm0.0}$ | $3.2_{\pm0.2}$ | $2.3_{\pm0.1}$ | 2.6 | $0.8_{\pm0.1}$ | $18.3_{\pm0.2}$ | $1.0_{\pm0.0}$ | $0.1_{\pm0.0}$ | 5.1 |
| ● EATA | $\mathbf{41.2}_{\pm0.1}$ | $\underline{25.0}_{\pm0.5}$ | $\mathbf{28.8}_{\pm0.6}$ | $\mathbf{32.3}_{\pm0.4}$ | $\underline{34.5}_{\pm0.2}$ | $\underline{33.2}_{\pm0.2}$ | $\underline{32.5}$ | $\underline{15.2}_{\pm0.4}$ | $\underline{29.2}_{\pm0.2}$ | $\underline{19.6}_{\pm0.3}$ | $5.8_{\pm0.1}$ | $\underline{17.4}$ |
| ● SAR | $10.9_{\pm0.7}$ | $2.1_{\pm0.2}$ | $1.0_{\pm0.0}$ | $2.0_{\pm0.0}$ | $3.2_{\pm0.2}$ | $2.3_{\pm0.1}$ | 3.6 | $1.1_{\pm0.0}$ | $19.8_{\pm0.1}$ | $1.5_{\pm0.0}$ | $0.3_{\pm0.0}$ | 5.7 |
| ● SoTTA | $13.8_{\pm0.6}$ | $10.1_{\pm0.3}$ | $8.4_{\pm0.2}$ | $4.2_{\pm0.2}$ | $6.4_{\pm0.2}$ | $3.4_{\pm0.1}$ | 7.7 | $2.4_{\pm0.1}$ | $20.4_{\pm0.1}$ | $4.4_{\pm0.1}$ | $1.1_{\pm0.0}$ | 7.1 |
| ● DeYO | $7.0_{\pm0.5}$ | $1.5_{\pm0.1}$ | $2.2_{\pm0.1}$ | $3.5_{\pm0.2}$ | $6.8_{\pm0.2}$ | $4.2_{\pm0.1}$ | 4.2 | $0.6_{\pm0.0}$ | $20.8_{\pm0.2}$ | $2.8_{\pm0.1}$ | $0.4_{\pm0.0}$ | 6.2 |
| ● CEMA | $6.8_{\pm0.3}$ | $1.9_{\pm0.1}$ | $2.0_{\pm0.1}$ | $2.9_{\pm0.2}$ | $4.4_{\pm0.2}$ | $3.9_{\pm0.1}$ | 3.7 | $0.9_{\pm0.0}$ | $19.9_{\pm0.1}$ | $1.9_{\pm0.1}$ | $0.1_{\pm0.0}$ | 5.7 |
| ● READ | $27.1_{\pm0.6}$ | $22.1_{\pm0.4}$ | $19.0_{\pm0.2}$ | $21.6_{\pm0.9}$ | $23.6_{\pm1.4}$ | $21.0_{\pm0.5}$ | 22.4 | $10.1_{\pm0.1}$ | $27.9_{\pm0.2}$ | $15.3_{\pm0.4}$ | $4.5_{\pm0.1}$ | 14.5 |
| ● SuMi | $\mathbf{41.9}_{\pm0.3}$ | $\mathbf{26.3}_{\pm0.2}$ | $\underline{27.9}_{\pm0.2}$ | $\underline{31.6}_{\pm0.3}$ | $\mathbf{37.1}_{\pm0.2}$ | $\mathbf{34.1}_{\pm0.1}$ | $\mathbf{33.2}$ | $\mathbf{18.4}_{\pm0.2}$ | $\mathbf{31.8}_{\pm0.2}$ | $\mathbf{21.7}_{\pm0.2}$ | $\mathbf{6.7}_{\pm0.1}$ | $\mathbf{19.7}$ |

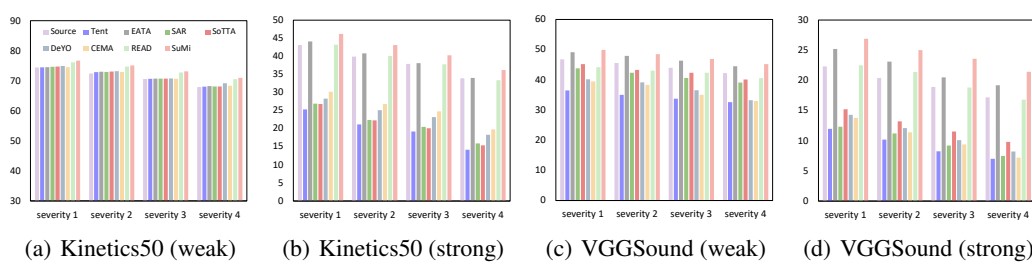

| (a) Kinetics50 (weak) | (b) Kinetics50 (strong) | (c) VGGSound (weak) | (d) VGGSound (strong) |

Figure 4: Comparison with SOTA methods on corrupted data of different severity levels. weak: average accuracy of 21 different types of weak OOD distribution shifts. strong: average accuracy of 4 different types of strong OOD distribution shifts.

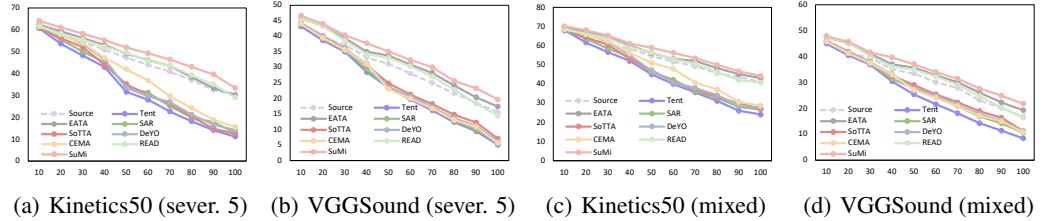

| (a) Kinetics50 (sever. 5) | (b) VGGSound (sever. 5) | (c) Kinetics50 (mixed) | (d) VGGSound (mixed) |

Figure 5: Comparison with SOTA methods on mixed corrupted data with ten different ratios of strong OOD samples. (a) and (b): severity level 5. (c) and (d): mixed severity.

batch size of 16/64 for Kinetics50-C and VGGSound-C, respectively. The multimodal threshold $\gamma_m$ in Equation 4 and the normalization factor Ent$_0$ in Equation 7 are set to $0.4 \times \ln C$ following Niu et al. (2022) by default where $C$ is the number of task classes. The unimodal threshold $\gamma_u$ in Equation 4 is set to $e^{-1}$ by default. The smoothing coefficient $\beta$ is set to 0.6/0.9, the weighting term $\lambda$ is set to 5.0 and the unimodal assistance $\mu$ is set to 1.0 by default for Kinetics50-C and VGGSound-C. For strong OOD adaptation, we set the mutual information sharing term $t_0$ as $iter/2$. Following previous work (Niu et al., 2023; Gong et al., 2023a; Chen et al., 2024; Guo et al., 2024b), we update the affine parameters of normalization layers.

## 4.2 COMPARISON WITH STATE-OF-THE-ARTS

We compare our method with Tent (Wang et al., 2021), EATA (Niu et al., 2022), SAR (Niu et al., 2023), SoTTA (Gong et al., 2023a), CEMA (Chen et al., 2024), DeYO (Lee et al., 2024) and READ (Yang et al., 2024).

**Single Domain Results.** We report the accuracy of 21 different types of weak OOD corruptions and 4 different types of strong OOD corruptions at severity level 5 on Kinetics50 and VGGSound in Table 1, 2, 3 and 4. For weak OOD samples, SuMi outperforms existing SOTA methods on most of the distribution shifts and achieves consistent good performances. For strong OOD samples, most of the existing SOTA methods perform even worse than the source model. On both datasets, only EATA performs better than the source model slightly. On the most noisy distribution "Mix" where

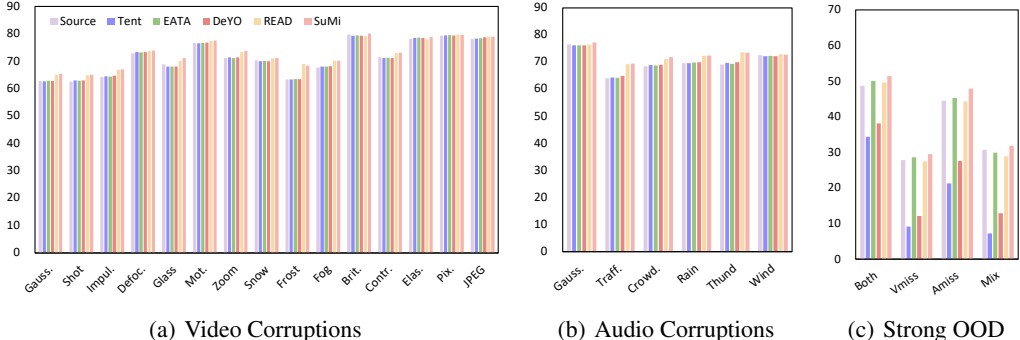

Figure 6: Comparison with SOTA methods on mixed severity level on Kinetics50-C.

one of the modality is missing and the other is corrupted, EATA also has a performance degradation, performing worse than the source model. In comparison, SuMi outperforms other methods consistently and significantly on all the four distribution scenarios, indicating its effectiveness and superiority in dealing with the complex noise patterns in multimodal data. Furthermore, we compare SuMi with SOTA methods at different severity levels and present the results in Figure 4. From the figure, we can observe that at different severity levels, most of the methods can work well on weak OOD samples while fail on strong OOD samples. However, SuMi can still perform well and achieve the best results on corrupted datasets at all the four severity levels, which demonstrates its generalization ability.

**Mixed Domain Results.** In Figure 5, we present the results of different methods on datasets with ten different portions of strong OOD samples. Figure 5(a) and 5(b) presents the results at severity level 5. We can observe that all the methods can perform well when the ratio of strong OOD samples is low. However, with the ratio increasing, the performance of most of the methods degrade rapidly, performing worse than the source model. The reason is that the huge distribution gap between the source domain and strong OOD domain destroy the prior knowledge of the source model, thus leading to a degradation of the model. In comparison, SuMi smooths the process by interquartile range smoothing and outperforms the SOTA methods consistently. From Figure 5(c) and 5(d) where mixed severity level cases are added, we can reach the same conclusion. Moreover, in Figure 6 and 10, we present the results on corrupted data with mixed severity level samples on both datasets. From the table, we can observe that on mixed severity level, SuMi can still achieve consistent improvements, outperforming other SOTA methods in most of the distribution shifts. Additionally, on strong OOD distribution shifts, other methods always fail while SuMi can still perform well. These results indicate the effectiveness of SuMi.

## 4.3 ABLATION STUDY

**Contributions of different components.** In Table 5, we present the results of our ablation experiments. We can observe that IQR smoothing brings the most improvements to the model. This is because IQR smoothing can bridge the gap between the source domain and strong domain, avoiding the abrupt distribution shifts which could destroy the prior knowledge from the source model. Unimodal assistance aims to select low-entropy samples with rich multimodal information for optimization and can also enhance the performance of the model. With these strategies combined, the performance of the model is further enhanced.

**Exploration of unimodal assistance.** In Equation 4 and Figure 2, we divide the samples into four areas. We can consider the four areas as Area 1 (low-entropy samples with rich multimodal information), Area 2 (high-entropy samples with rich multimodal information), Area 3 (low-entropy samples with little multimodal information) and Area 4 (high-entropy samples with little multimodal information). We present the performance of these four areas on Kinetics50-C in Table 6. From the table, we can observe that selecting low-entropy samples for optimization will yield better results. Based on low-entropy samples, rich multimodal information will further help to optimize the multimodal models and achieve better results.

Table 5: Ablation study of different components in SuMi on corrupted data with 50% of strong OOD samples at different severity levels. IQR, UA and MIS represents IQR smoothing, unimodal assistance and mutual information sharing, respectively.

| | | | Kinetics50-C | | | VGGSound-C | | |
|---|---|---|---|---|---|---|---|---|
| IQR | UA | MIS | severity 3 | severity 5 | mixed severity | severity 3 | severity 5 | mixed severity |
| | | | 37.1 | 31.7 | 36.4 | 25.7 | 23.5 | 25.3 |
| ✓ | | | 52.1 | 45.1 | 51.9 | 33.8 | 30.4 | 33.1 |
| | ✓ | | 49.4 | 39.4 | 46.2 | 31.1 | 27.4 | 31.2 |
| | | ✓ | 47.4 | 38.1 | 45.6 | 29.8 | 26.1 | 28.4 |
| ✓ | ✓ | | 58.0 | 51.2 | 57.4 | 36.9 | 34.3 | 36.5 |
| ✓ | | ✓ | 56.0 | 49.7 | 56.7 | 34.2 | 32.1 | 34.0 |
| | ✓ | ✓ | 54.3 | 44.6 | 51.3 | 33.4 | 29.8 | 32.1 |
| ✓ | ✓ | ✓ | 59.3 | 52.0 | 59.1 | 38.4 | 35.1 | 38.3 |

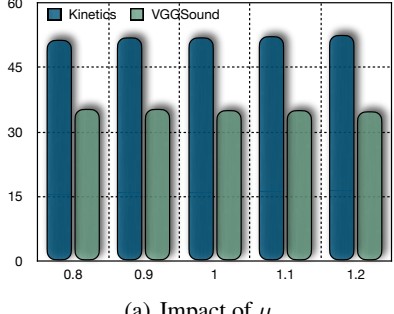

(a) Impact of $\mu$

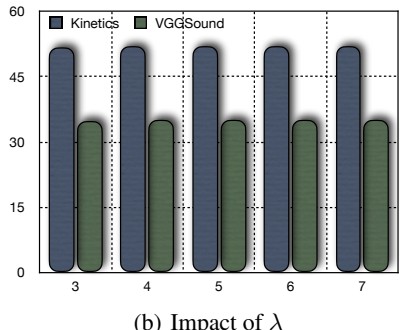

(b) Impact of $\lambda$

Figure 7: Ablation experiments of hyperparameters on the two datasets.

Additionally, we explore the trade-off coefficient $\mu$ in Equation 4 and present the results on both datasets in Figure 7(a). From the figure, we can observe that with the increase of $\mu$, the performance improves on Kinetics50-C and drops on VGGSound-C. This is because Kinetics50 is a video modality dominant dataset and VGGSound is an audio modality dominant dataset. From the results, we know that adding more weight to the dominant modality will yield poorer performance because unimodal assistance aims to select samples with rich multimodal information. Therefore, adding weight to the weak modality will help to utilize the multimodal features. Besides, the performances with different $\mu$ are stable, indicating the stability of the strategy.

Table 6: Performance of samples in different areas on Kinetics50-C.

| | Ratios | Acc |
|---|---|---|
| Area 1 | 13.1% | 39.4 |
| Area 2 | 78.5% | 27.6 |
| Area 3 | 5.3% | 32.1 |
| Area 4 | 3.1% | 24.3 |

**Exploration of $\lambda$.** To explore the mutual information sharing, we select several $\lambda$ in Equation 8 and present the results on both datasets in Figure 7(b). We can observe that increasing the weight term $\lambda$ will improve the performance slightly. Besides, the results in the table demonstrate the stability of mutual information sharing across varying values of $\lambda$.

## 5 CONCLUSION

In this paper, we propose a new practical and challenging task named multimodal wild TTA. To address this problem, we propose sample identification with interquartile range smoothing and unimodal assistance, and mutual information sharing (SuMi). SuMi bridges the gap between the source domain and strong OOD domain by smoothing the adaptation using interquartile range. Besides, SuMi leverages unimodal features to select low-entropy samples with rich multimodal information for optimization. Finally, mutual information sharing is proposed to further align the information and reduce the discrepancies across different modalities. We conduct extensive experiments on two widely used multimodal datasets where SuMi outperforms existing TTA methods significantly and consistently, indicating its effectiveness. Ablation experiments are then conducted to validate the contributions of each component.

ACKNOWLEDGEMENT

This work was supported in part by Public Welfare Research Program of Ningbo under Grant No. 2024S062 and Yongjiang Talent Project of Ningbo under Grant No. 2024A-161-G.

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

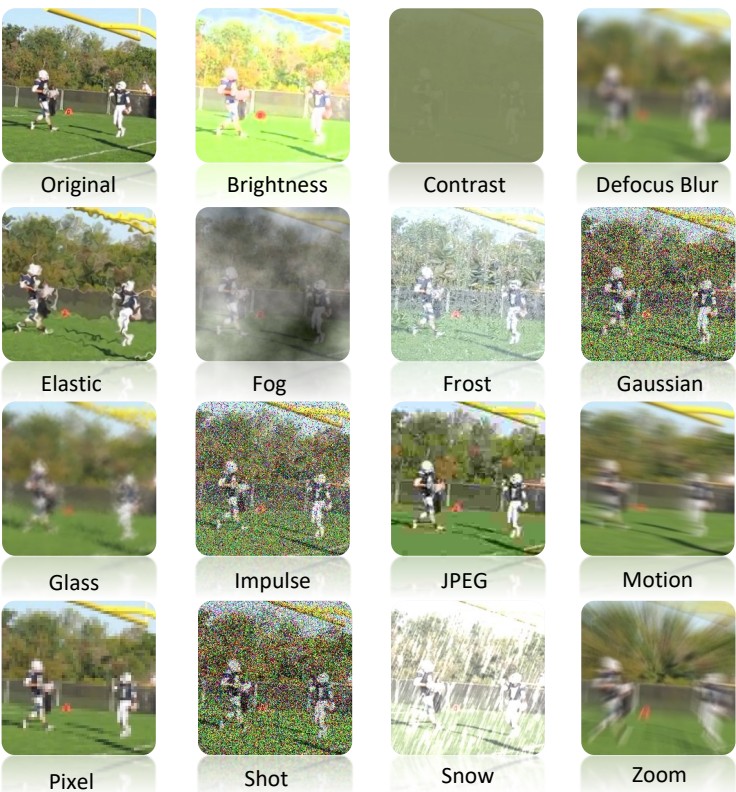

Figure 8: Fifteen different types of noises in videos.

## A    DETAILS OF DATASETS

**Kinetics50** (Kay et al., 2017). The Kinetics dataset is a large-scale and high-quality dataset for human action recognition in videos. The dataset consists of around 500,000 video clips covering 600 human action classes with at least 600 video clips for each action class. Each video clip lasts around 10 seconds and is labeled with a single action class. The videos are collected from YouTube. Following Yang et al. (2024), we use a subset of Kinetics which consists of 50 classes, 29,204 training pairs and 2,466 test pairs.

**VGGSound** (Chen et al., 2020). VGGSound is a large-scale audio-visual correspondent dataset consisting of short clips of audio sounds, extracted from videos uploaded to YouTube. All videos are captured "in the wild" with audio-visual correspondence in the sense that the sound source is visually evident. Each video in this dataset has a fixed duration of 10 seconds.

To evaluate the performance under different distribution shifts, we introduce a total of 25 different types of distribution shifts. These distribution shifts can be divided into two groups: weak OOD distribution shifts and strong OOD distribution shifts.

For weak distribution shifts, we divide them into video corruptions and audio corruptions. Following previous work (Hendrycks & Dietterich, 2019), we introduce 15 different types of video corruptions as shown in Figure 8. They include "Gaussian Noise" (Gauss.), "Shot Noise" (Shot), "Impulse Noise" (Impul.), "Defocus Blur" (Defoc.), "Glass Blur" (Glass), "Motion Blur" (Motion), "Zoom Blur" (Zoom), "Snow" (Snow), "Frost" (Frost), "Fog" (Fog), "Brightness" (Brit.), "Contrastive" (Contr.), "Elastic" (Elastic), "Pixelate" (Pixel) and "JPEG" (JPEG). Following Yang et al. (2024), we introduce six types of audio corruptions as shown in Figure 9. They include "Gaussian Noise" (Gauss.), "Paris Traffic Noise" (Traff.), "Crowd Noise" (Crowd), "Rainy Noise" (Rain), "Thunder Noise" (Thund.) and "Windy Noise" (Wind).

For strong distribution shifts, in this paper, we introduce four types of corruptions. They include "Both Modality Corruptions" (Both), "Audio Missing" (Amiss), "Video Missing" (Vmiss) and

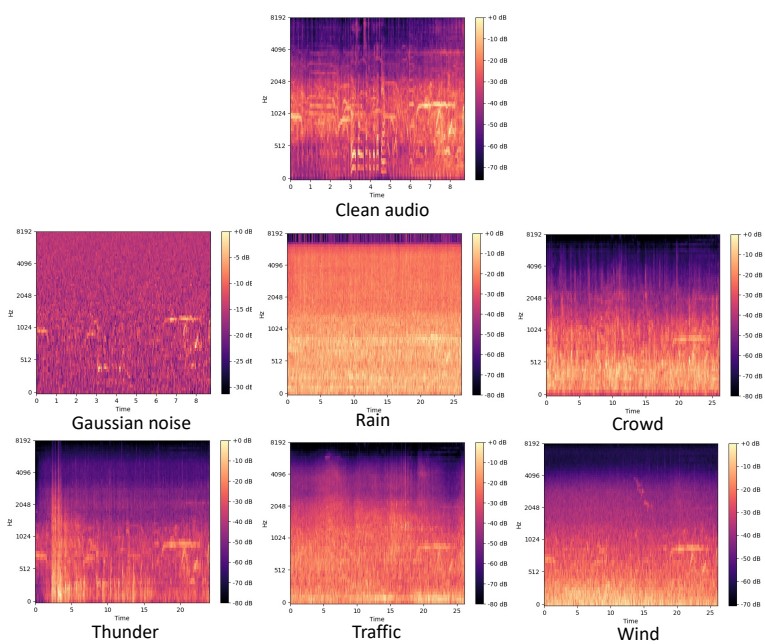

Figure 9: Six different types of noises in audio.

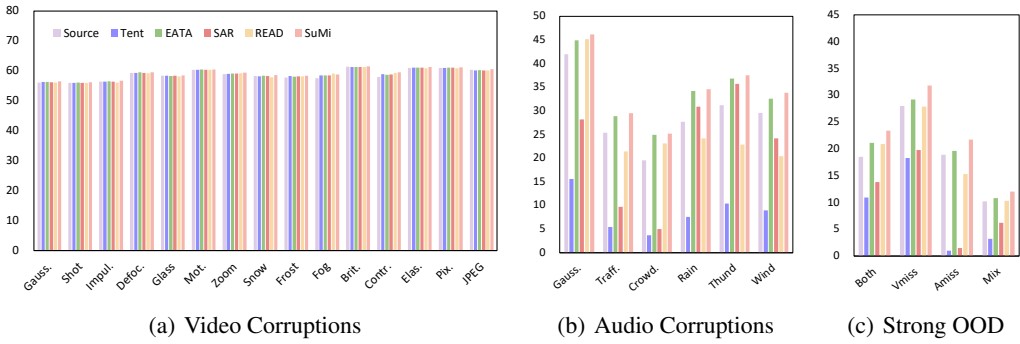

(a) Video Corruptions    (b) Audio Corruptions    (c) Strong OOD

Figure 10: Comparison with SOTA methods on mixed severity level on VGGSound-C.

"Missing and Corruption" (Mix). Both represents the both modalities are corrupted. Mix represents that one of the modality is missing and the other is corrupted. For missing modality, we substitute any missing modalities with zero vectors. This allows us to maintain the input dimensions required by the network while enabling it to process the available data effectively.

## B  DETAILS OF METHOD AND IMPLEMENTATION

IQR can effectively capture the central tendency and variability of the data. For IQR calculation, we can use different metrics for data ranking such as magnitude (Euclidean Norm) and specific dimension comparison. However, there are some drawbacks of these metrics. For example, large components in the vectors will disproportionately affect the magnitude. Specific dimension comparison ignores other dimensions which may be important and does not represent the overall vector well. To combine multiple dimensions, we calculate the min and max of all the $h$ element by element to obtain $h_{min}$ and $h_{max}$. Then, we obtain the $Q_1$ and $Q_3$ through linear interpolation. Additionally, we follow previous work (Yang et al., 2024) and add a negative entropy loss term to balance the prediction.

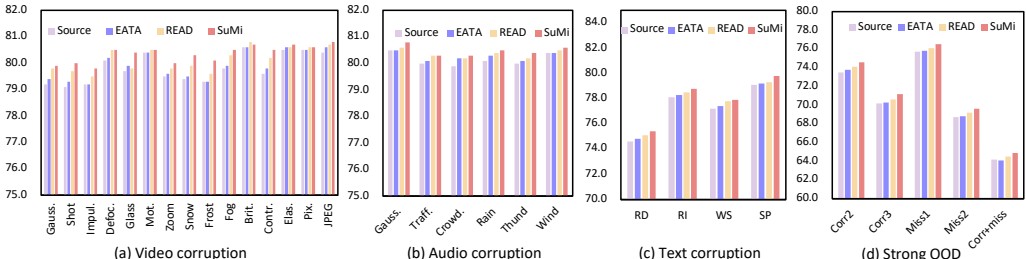

Figure 11: Performance comparisons on CMU-MOSI.

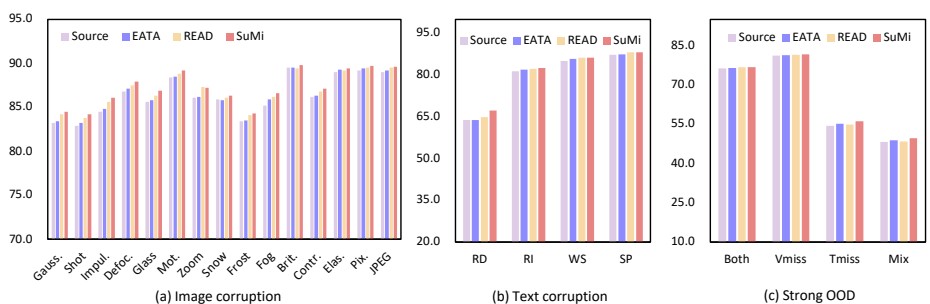

Figure 12: Performance comparisons on UPMC-Food 101.

# C MORE EXPERIMENTAL RESULTS

## C.1 GENERALIZATION ABILITY OF SUMI

To further validate the generalization ability and robustness of SuMi, we conduct experiments on two additional datasets. The first dataset is CMU-MOSI (Zadeh et al., 2016). MOSI encompasses three modalities (text, image, and audio), enabling us to evaluate the performance of SuMi across a dataset with more than two modalities. The other dataet is UPMC-Food101 (Wang et al., 2015), which is a image-text dataset for food classification.

We introduce four types of corruptions for text modality. Specifically, we introduce random deletion of word or character (RD), random insertion of word or character (RI), word shuffling (WS) and sentence permutation (SP). Random deletion of word or character randomly removes words or characters from sentences to simulate noise in the data. Random insertion inserts random words or characters into sentences, which can disrupt the original meaning. Word shuffling randomly shuffle words within a sentence to change the sentence structure while retaining some semantic meaning. Sentence permutation changes the order of sentences in a paragraph to simulate context shifts. For strong OOD on MOSI, we use Corrn to denote that n modalities are corrupted, missn to denote n modalities are missing and corr+miss to denote both missing modalities and corruption modalities are present.

We use the stacked transformer blocks trained on MOSI dataset. Then, we fine-tune the model on corrupted MOSI. For Food 101, we use the CLIP image encoder and text encoder as the modality-specific encoders followed by a fusion classification head. Then we train the model on Food 101 and adapt the model on the corrupted Food 101. The results are presented in Figure 11 and Figure 12. We can observe that in dataset with more modalities and in the common vision-language task, SuMi can also outperform existing methods, demonstrating its effectiveness.

Table 7: Results on real-world distribution shifts.

| Method | MOSI→ SIMS | | SIMS→ MOSI | |
|--------|------|------|------|------|
| | ACC | F1 | ACC | F1 |
| Source | 39.2 | 39.1 | 40.1 | 45.5 |
| EATA | 40.5 | 41.2 | 40.4 | 45.7 |
| READ | 42.0 | 42.5 | 40.9 | 46.9 |
| SuMi | 44.2 | 44.7 | 41.6 | 47.8 |

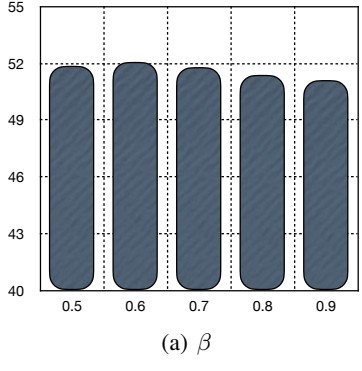

(a) $\beta$

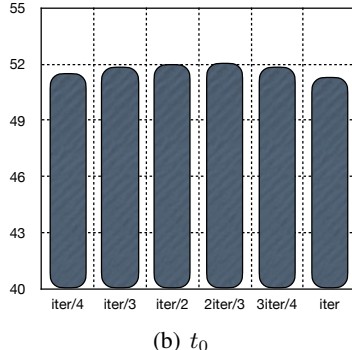

(b) $t_0$

Figure 13: Ablation experiments on $\beta$ and $t_0$ on Kinetics50-C.

## C.2 REAL-WORLD DISTRIBUTION SHIFTS

To evaluate the robustness of SuMi in addressing real-world distribution shift, we follow the setting of Guo et al. (2024b) and conduct experiments on two datasets: CMU-MOSI (Zadeh et al., 2016) and CH-SIMS (Yu et al., 2020). Specifically, CMU-MOSI and CH-SIMS are multimodal sentiment analysis datasets which include three modalities. They contain different topics of conversations, different speakers, and different recording environments which can all be seen as real-world distribution shifts. In pratice, we consider the task as a binary classification task and use the cross entropy loss. We use stacked Transformer blocks as the backbone and pre-train the model on CMU-MOSI and CH-SIMS as the source model for the setting MOSI→ SIMS and SIMS→ MOSI, respectively. Table 7 presents the results. We can observe that in real-world distribution shifts, SuMi can still outperform existing methods, showing its robustness.

## C.3 MORE ABLATION EXPERIMENTS

**Exploration of $\beta$ in IQR smoothing.** In interquartile range smoothing, we set $\beta$ for more stable selection. Here, we select different values of $\beta$ and present the results in Figure 13(a). From the figure, we can observe that the performances across varying $\beta$ are stable.

**Exploration of $t_0$ in mutual information sharing.** For strong OOD adaptation, we add mutual information sharing in the first $t_0$ iterations to avoid the impact of strong OOD samples which could damage the mutual information sharing. We select several different $t_0$ to conduct experiments and present the results on Kinetics50-C in Figure 13(b). From the figure, we can observe that the performances are all better than the model without mutual information sharing which indicates the effectiveness of mutual information sharing strategy. Besides, with the increase of $t_0$, the performance improves before dropping when $t_0 = \frac{3iter}{4}$. This shows that with the adaptation process, the strong OOD samples also increase which could bring many noises and damage the mutual information sharing. Moreover, the performances across varying $t_0$ are stable, demonstrating the stability of our method.

**Exploration of smoothing process.** In Equation 3, we opt for a simple linear smoothing process for clarity. Here, we provide a deeper analysis of the smoothing process. In addition to the linear smoothing, we provide the results of the logarithmic and exponential functions. Specifically, for logarithmic function, we use $f(t) = \log(\frac{(e-1)t}{iter} + 1)$ and for exponential function we use $f(t) =$

Table 8: Performance with different smoothing functions on Kinetics50-C.

| $f(t)$ | Linear | Exponential | Logarithmic |
|---|---|---|---|
| Acc | 59.1 | 59.5 | 58.7 |

$\exp\left(\frac{t \ln 2}{iter}\right) - 1$. We present the results in Table 8. From the table, we can observe that using $f(t) = \exp\left(\frac{t \ln 2}{iter}\right) - 1$ function can improve the performance of the model slightly. From the properties of the exponential function, it can be seen that the function grows slowly when the variable $t$ is small and quickly when the variable is large. For logarithmic function, it grows quickly when the variable $t$ is small and slowly when the variable is large. This indicates that slowing down the smoothing process in the initial phase helps the model's performance. Additionally, we can observe that the function will not affect the performance drastically, indicating the effectiveness of smoothing process itself.

