# OpenReview forum: "Smoothing the Shift: Towards Stable Test-Time Adaptation under Complex Multimodal Noises"
_ICLR.cc/2025/Conference — ICLR 2025 Poster_

### Official Review · Reviewer_Xmz9 · 2024-10-24

**Soundness:** 3
**Presentation:** 2
**Contribution:** 2
**Rating:** 6
**Confidence:** 3

**Summary:**

This paper mainly focus on the task of test-time adaptation. Different from previous works, the author consider a much tougher circumstance with both weak and strong OOD samples. Performance of existing methods will inevitably degrade for huge distribution gap. To combat with this gap, the authors introduce the interquartile to smooth the sharp gap. Moreover, unimodal entropy is utilized to suggest rich multimodal information. Since strong OOD may contain missing modality problem, the author further a mutual information sharing strategy to complement the distribution. Experiments on several datasets under various domains demonstrate the performance of the proposed method.

**Strengths:**

1. The paper is written straight-forward, which is easy to follow.
2. The researched TTA problem is a popular problem, while the authors make a further step to discuss under both weak and strong OOD situations.

**Weaknesses:**

1. While I understand the general framework of the method, certain parts seem to lack clarity. For instance, in section 3.2.1, I can learn that the purpose is to select those samples with lower adaptation gap, however, the ranking process for samples is never introduced. In equation 3, vector $h$ is directly used for comparison, but the comparison method—whether by calculating magnitude, norm, or another metric—is not specified. Definition of IQR also lacks clear clarification. Besides, the motivation part (Figure 3) is suggested to be placed in the Intro section, since it's more vivid and persuasive than experiment results (Figure 1).
2. My main concern about this work is novelty. The method is mainly composed of three parts: interquartile mainly smooth the distribution gap, unimodal entropy selects samples with high quality, and mutual information sharing combats with missing modality. Every part seems to be orthogonal, with no clear interaction between them. Please explain how separate parts benefit each other in detail, or does every part just simply make their own contribution?
3. While I am mostly ok with the experimental results, I have concern about the setting of the experiment. In my view, missing modality is a more common issue in multimodal circumstances with more than two modalities. Discussing such problem only on simulated datasets with two modalities is somehow narrow. It is suggested for the author to conduct experiments on dataset with more than two modalities.

**Questions:**

Please refer to cons above.

---

> ### Author Response · Authors · 2024-11-16
> **Rebuttal (1/2)**
>
> Thank you for your detailed review and your recognition of our work. We appreciate your insightful comments and suggestions. Here are our responses.
>
> **Q1: Method clarification.**
>
> Thank you for your feedback. For ranking process, during the method design phase, we have considered several metrics such as magnitude (Euclidean Norm) and specific dimension comparison. However, there are some drawbacks of these metrics. For example, large components in the vectors will disproportionately affect the magnitude. Specific dimension comparison ignores other dimensions which may be important and does not represent the overall vector well. Therefore, to combine multiple dimensions, we calculate the min and max of all the $h$ element by element to obtain $h_{min}$ and $h_{max}$. Then, we obtain the Q1 and Q3 through linear interpolation. Specifically, Q1 is the 25th percentile and Q3 is the 75th percentile. Consequently, from the experiments, we find that we can achieve better and more robust results than using magnitude or specific dimension comparison.
>
> In our sample identification process, $h$ is a vector and the lower bound $Q_1-\frac{3}{2}IQR$ and the upper bound $Q_3+\frac{3}{2}IQR$ are also vectors which have the same dimension as $h$. We compare these two vectors element by element. As we illustrate in the paper, we select $h$ for adaptation if $\beta+\frac{(1-\beta)t}{iter}$ percent of the values in $h$ fall in the range [$Q_1-\frac{3}{2}IQR$, $Q_3+\frac{3}{2}IQR$] for stability. And we also explore the $\beta$ in our ablation experiments.
>
> In case of the definition of IQR, we have given the full definition and calculation of IQR in Definition 1. For clarity, we have revised the manuscript to present a better understanding of calculating the IQR and quartiles, including the ranking criteria in the appendix in detail. For the fig3, we have revised the manuscript and cited it in our introduction section. Thank you for your suggestions.
>
>
>
> **Q2: Relationship between three strategies.**
>
> We would like to clarify that the three strategies we proposed are interconnected and not independent of each other. First, IQR is used to smooth the adaptation process to avoid abrupt huge distribution gap. IQR smoothing can help unimodal assistance identify the rich multimodal data more accurately. In the first few iterations, without IQR smoothing, it is hard for unimodal assistance to identify high-confident samples with rich information because the batch contains many strong OOD samples where the source model cannot handle well. Therefore, based on IQR smoothing, the function of unimodal assistance will be boosted, especially in mixed OOD samples. Meanwhile, IQR smoothing and unimodal assistance deal with the shift from two perspectives. IQR smoothes the adaptation process and unimodal assistance selects the data which are good for the entropy minimization. Additionally, mutual information sharing is introduced to reduce the discrepancies across different modalities and enhance the information utilization of different modalities. This is beneficial to unimodal assistance because mutual information sharing can help the unimodal branch produce more confident unimodal predictions, thus enhancing the unimodal assistance strategy. Therefore, each component is designed to complement the others, enhancing the overall effectiveness of our approach. And we can observe from the ablation experiments that the strategies combined can further improve the performance of SuMi.

---

> ### Author Response · Authors · 2024-11-16
> **Rebuttal (2/2)**
>
> **Q3: Effectiveness of the method across multiple modalities.**
>
> Thank you for your suggestions. To evaluate the performance of SuMi on additional datasets, we select MOSI as the dataset. We choose CMU-MOSI for two primary reasons. First, it includes a text modality, allowing us to work with datasets that extend beyond just video and audio. Second, MOSI encompasses three modalities—text, image, and audio—enabling us to evaluate the performance of SuMi across a dataset with more than two modalities.
>
> Here, we introduce four types of corruptions for text modality. Specifically, we introduce random deletion of word or character (RD), random insertion of word or character (RI), word shuffling (WS) and sentence permutation (SP). We have added detailed explanations for these corruptions to the appendix of our manuscript. For backbone, we use the stacked transformer blocks trained on MOSI dataset. Then, we fine-tune the model on corrupted MOSI. Here are the results.
>
> | Method | Gauss. | Shot | Impul. | Defoc. | Glass | Mot. | Zoom | Snow | Frost | Fog  | Brit. | Contr. | Elas. | Pix. | JPEG |
> | ------ | ------ | ---- | ------ | ------ | ----- | ---- | ---- | ---- | ----- | ---- | ----- | ------ | ----- | ---- | ---- |
> | Source | 79.2   | 79.1 | 79.2   | 80.1   | 79.7  | 80.4 | 79.5 | 79.4 | 79.3  | 79.8 | 80.6  | 79.6   | 80.5  | 80.5 | 80.4 |
> | EATA   | 79.4   | 79.3 | 79.2   | 80.2   | 79.9  | 80.4 | 79.6 | 79.5 | 79.3  | 79.9 | 80.6  | 79.8   | 80.6  | 80.5 | 80.6 |
> | READ   | 79.8   | 79.7 | 79.5   | 80.5   | 79.8  | 80.5 | 79.8 | 79.9 | 79.6  | 80.3 | 80.8  | 80.2   | 80.6  | 80.6 | 80.7 |
> | SuMi   | 79.9   | 80.0 | 79.8   | 80.5   | 80.4  | 80.5 | 80.0 | 80.3 | 80.1  | 80.5 | 80.7  | 80.5   | 80.7  | 80.6 | 80.8 |
>
> | Method | Gauss. | Traff. | Crowd. | Rain | Thund | Wind |
> | ------ | ------ | ------ | ------ | ---- | ----- | ---- |
> | Source | 80.5   | 80.0   | 79.9   | 80.1 | 80.0  | 80.4 |
> | EATA   | 80.5   | 80.1   | 80.2   | 80.3 | 80.1  | 80.4 |
> | READ   | 80.6   | 80.3   | 80.2   | 80.4 | 80.2  | 80.5 |
> | SuMi   | 80.8   | 80.3   | 80.3   | 80.5 | 80.4  | 80.6 |
>
> | Method | RD   | RI   | WS   | SP   |
> | ------ | ---- | ---- | ---- | ---- |
> | Source | 74.6 | 78.1 | 77.2 | 79.1 |
> | EATA   | 74.8 | 78.3 | 77.4 | 79.2 |
> | READ   | 75.1 | 78.5 | 77.8 | 79.3 |
> | SuMi   | 75.4 | 78.8 | 77.9 | 79.8 |
>
> | Method | Corr2 | Corr3 | Miss1 | Miss2 | Corr+miss |
> | ------ | ----- | ----- | ----- | ----- | --------- |
> | Source | 73.5  | 70.2  | 75.7  | 68.7  | 64.2      |
> | EATA   | 73.8  | 70.3  | 75.8  | 68.8  | 64.1      |
> | READ   | 74.1  | 70.6  | 76.1  | 69.2  | 64.5      |
> | SuMi   | 74.6  | 71.2  | 76.5  | 69.6  | 64.9      |
>
> For strong OOD, we use Corrn to denote that n modalities are corrupted, miss n to denote n modalities are missing and corr+miss to denote both missing modalities and corruption modalities are present. From the table, we can observe that in dataset with more modalities, SuMi can also outperform existing method, demonstrating its effectiveness. We have added these results in our revised manuscript.
>
> We have made revisions to our paper in response to your suggestions. Specific details can be found in the updated manuscript. If you have further questions, please feel free to reach out.

---

> ### Comment · Reviewer_Xmz9 · 2024-11-17
>
> Thank you for your detailed response. I'll keep my score.

---

### Official Review · Reviewer_pfaw · 2024-10-26

**Soundness:** 3
**Presentation:** 3
**Contribution:** 3
**Rating:** 6
**Confidence:** 5

**Summary:**

This paper investigates multimodal test-time adaptation (TTA) and introduces a new method, SuMI. Specifically, it extends current multimodal TTA research into a more challenging setting, termed wild multimodal TTA, where test data may include weak or strong out-of-distribution (OOD) contamination. To tackle this problem, the authors propose a technically robust approach involving reliable data selection and cross-modal discrepancy elimination. Extensive experimental results confirm the effectiveness of the proposed method.

**Strengths:**

Given the prevalence of multimodal data in the real world, studying multimodal test-time adaptation (TTA) methods for pre-trained models is highly relevant, especially in the era of foundation models. Beyond existing multimodal research, such as READ (Yang et al., 2024), the authors extend this work into a more challenging and general setting—wild multimodal TTA. I believe this extension could significantly advance test-time adaptation research within the community.

**Weaknesses:**

1. Figure 2 needs revision; it’s currently unclear what is being input to the fusion layers.
2. Figure 3(c) lacks clear explanations for the X and Y axes.
3. The settings (weak/strong OOD) should be labeled in Tables 1 and 3.
4. The variable $\mu$ is not found in Equation 4. Did you mean $\gamma_{\mu}$?
5. It’s unclear how the proposed method handles cases where certain modalities are missing. From my understanding, if one modality is missing, the correspondence predictions may not be achievable.

**Questions:**

Please see the weaknesses

---

> ### Author Response · Authors · 2024-11-16
> **Rebuttal**
>
> Thank you for your detailed review and your recognition of our work. We appreciate your insightful comments and suggestions. Here are our responses.
>
> **Q1: Figure 2 needs revision; it’s currently unclear what is being input to the fusion layers.**
>
> Sorry for the confusion. In figure 2, we input the filtered representations by IQR smoothing into the fusion layers. Specifically, after the encoders, we will obtain representations of each modality. Then, we select the representations using IQR smoothing and input these selected features into the fusion layers. We input the features into the same fusion layer three times to get multimodal prediction and unimodal predictions. Here we present the pseudo-code for a better illustration:
>
> ```
> # afeat and vfeat are the features of audio and video modalities selelcted by IQR smoothing
> aoutput = fusion(x=afeat)
> voutput = fusion(x=vfeat)
> feats = cat[afeat, vfeat]
> outputs = fusion(x=feats)
> ```
>
> We have revised the figure in our paper for a better presentation.
>
>
>
> **Q2: Figure 3(c) lacks clear explanations for the X and Y axes.**
>
> We would like to clarify that the labels for the X and Y axes are included in the figure caption. However, we understand that they may not have been sufficiently clear. We have revised the figure and its caption to enhance clarity and ensure that the axes are easily interpretable for all readers.
>
>
>
> **Q3: The settings (weak/strong OOD) should be labeled in Tables 1 and 3.**
>
> Thank you for your feedback. Table 1 and 3 present the results of data with corrupted video modality (i.e. only weak OOD data of video modality). Table 2 and 4 include the strong OOD (multiple modalities corrupted) and have been labeled in the tables.
>
>
>
> **Q4: $\mu$ is not in Equation 4.**
>
> We would like to clarify that $\mu$ is indeed included in equation 4.  We've double-checked the equation and can confirm that $\mu$ is correctly included. It is a trade-off between the two modalities.
>
>
>
> **Q5: How to obtain the predictions when on modality is missing?**
>
> In our approach, we follow established practices from previous research by substituting any missing modalities with zero vectors. This allows us to maintain the input dimensions required by the network while enabling it to process the available data effectively. We have clarified this methodology in the revised manuscript to ensure that it is clearly articulated.
>
>
>
> We have made revisions to our paper in response to your suggestions. Specific details can be found in the updated manuscript. If you have further questions, please feel free to reach out.

---

> > ### Comment · Reviewer_pfaw · 2024-11-26
> >
> > Thank you for the response, I will maintain my positive rating.

---

### Official Review · Reviewer_d57C · 2024-10-30

**Soundness:** 2
**Presentation:** 3
**Contribution:** 2
**Rating:** 5
**Confidence:** 5

**Summary:**

This paper introduces a novel task called multimodal wild TTA and presents a method named Sample Identification with Interquartile Range Smoothing and Unimodal Assistance and Mutual Information Sharing (SuMi). SuMi addresses the challenges of bridging the source domain and out-of-distribution scenarios through interquartile range smoothing. It effectively selects low-entropy samples rich in multimodal information for optimization and employs mutual information sharing to align different modalities. Experiments on two popular multimodal datasets show that SuMi significantly outperforms existing TTA methods, confirming its effectiveness.

**Strengths:**

The author proposes a TTA task that seems to be more applicable, which is helpful for the subsequent development of TTA. The paper is well-motivated and easy to follow. And the author's related work review is relatively comprehensive. From the experiments, the proposed SuMi has achieved significant performance improvement in new TTA task.

**Weaknesses:**

1. My main concern is the motivation behind the paper. Is this innovative task based on a pseudo-motivation? Is it truly feasible for this task to occur in real-world application scenarios? See the Questions part for details.
2. Under real conditions of TTA, the number of modalities is generally more than two. I noticed that the authors discuss a multimodal form in equation (5), but at other times they refer only to a two-modality form. Can these be expanded into a general multimodal setting? Is it possible to validate the effectiveness of the method across multiple modalities in the experiments?
3. The authors provide a comprehensive discussion of the parameters. However, there is no explanation for why the 'Interquartile' form of IQR was chosen. In this specific task, other quantile ranges might yield better results. Additionally, the paper's improvement is limited to gradually smoothing with the number of iterations, which is merely a simple linear process. There is no discussion of a more theoretical or effective smoothing process.
4. The paper mentions that the selection of hyperparameters (such as $\mu$) relies on prior information about the dominant modality of the datasets. However, in TTA, we cannot know the prior information of the data. Could this directly impact the performance of the method across different datasets? Is there a more robust hyperparameter-selection mechanism?

**Questions:**

I would suggest that the authors consider discussing the setting of the new task and the practical applications of its motivation in detail. In real-world scenarios, would algorithms truly continue to be utilized under conditions where sensor failures or malfunctions occur? From my perspective, this task appears to resemble a combination of a modality missing and a TTA subtask. The proposed approach seems to specifically address these two subtasks.

---

> ### Author Response · Authors · 2024-11-16
> **Rebuttal (1/2)**
>
> Thank you for your detailed and constructive comments on our paper. We appreciate your insightful comments and suggestions. Here are our responses.
>
> **Q1: Discussing the practical applications of the method in detail.**
>
> We would like to clarify that the scenarios we address are indeed common in real-world applications.
>
> Let us first illustrate weak OOD and strong OOD samples in multimodal data. Weak OOD samples are those with only one modality corrupted. Strong OOD samples are those with multiple samples corrupted or missing modality. Strong OOD samples are quite common in real-world applications. For instance, consider a vehicle operating during a thunderstorm: the camera may capture visual information affected by raindrops, leading to noisy visual data, while the audio sensors are also impacted by the sounds of thunder and rain, resulting in noisy audio signals. Additionally, situations where a sensor fails due to weather conditions or external factors can lead to one modality being missing while another is corrupted. These strong out-of-distribution (OOD) conditions, where multiple modalities are affected by environmental factors, are quite prevalent. This illustrates the practical relevance of our task and underscores the importance of addressing both missing modalities and sensor disturbances in a cohesive framework, especially at test time. Furthermore, our intent is to handle distribution shifts in real-world applications. Missing modality is a common issue in multimodal noises. Meanwhile, different from existing methods to address missing modalities at train time with the labels, our method address it at test time without labels.
>
>
>
> **Q2: Effectiveness of the method across multiple modalities.**
>
> Thank you for your suggestions. To evaluate the performance of SuMi on additional datasets, we select MOSI as the dataset. We choose CMU-MOSI for two primary reasons. First, it includes a text modality, allowing us to work with datasets that extend beyond just video and audio. Second, MOSI encompasses three modalities—text, image, and audio—enabling us to evaluate the performance of SuMi across a dataset with more than two modalities.
>
> Here, we introduce four types of corruptions for text modality. Specifically, we introduce random deletion of word or character (RD), random insertion of word or character (RI), word shuffling (WS) and sentence permutation (SP). We have added detailed explanations for these corruptions to the appendix of our manuscript. For backbone, we use the stacked transformer blocks trained on MOSI dataset. Then, we fine-tune the model on corrupted MOSI. Here are the results.
>
> | Method | Gauss. | Shot | Impul. | Defoc. | Glass | Mot. | Zoom | Snow | Frost | Fog  | Brit. | Contr. | Elas. | Pix. | JPEG |
> | ------ | ------ | ---- | ------ | ------ | ----- | ---- | ---- | ---- | ----- | ---- | ----- | ------ | ----- | ---- | ---- |
> | Source | 79.2   | 79.1 | 79.2   | 80.1   | 79.7  | 80.4 | 79.5 | 79.4 | 79.3  | 79.8 | 80.6  | 79.6   | 80.5  | 80.5 | 80.4 |
> | EATA   | 79.4   | 79.3 | 79.2   | 80.2   | 79.9  | 80.4 | 79.6 | 79.5 | 79.3  | 79.9 | 80.6  | 79.8   | 80.6  | 80.5 | 80.6 |
> | READ   | 79.8   | 79.7 | 79.5   | 80.5   | 79.8  | 80.5 | 79.8 | 79.9 | 79.6  | 80.3 | 80.8  | 80.2   | 80.6  | 80.6 | 80.7 |
> | SuMi   | 79.9   | 80.0 | 79.8   | 80.5   | 80.4  | 80.5 | 80.0 | 80.3 | 80.1  | 80.5 | 80.7  | 80.5   | 80.7  | 80.6 | 80.8 |
>
> | Method | Gauss. | Traff. | Crowd. | Rain | Thund | Wind |
> | ------ | ------ | ------ | ------ | ---- | ----- | ---- |
> | Source | 80.5   | 80.0   | 79.9   | 80.1 | 80.0  | 80.4 |
> | EATA   | 80.5   | 80.1   | 80.2   | 80.3 | 80.1  | 80.4 |
> | READ   | 80.6   | 80.3   | 80.2   | 80.4 | 80.2  | 80.5 |
> | SuMi   | 80.8   | 80.3   | 80.3   | 80.5 | 80.4  | 80.6 |
>
> | Method | RD   | RI   | WS   | SP   |
> | ------ | ---- | ---- | ---- | ---- |
> | Source | 74.6 | 78.1 | 77.2 | 79.1 |
> | EATA   | 74.8 | 78.3 | 77.4 | 79.2 |
> | READ   | 75.1 | 78.5 | 77.8 | 79.3 |
> | SuMi   | 75.4 | 78.8 | 77.9 | 79.8 |
>
> | Method | Corr2 | Corr3 | Miss1 | Miss2 | Corr+miss |
> | ------ | ----- | ----- | ----- | ----- | --------- |
> | Source | 73.5  | 70.2  | 75.7  | 68.7  | 64.2      |
> | EATA   | 73.8  | 70.3  | 75.8  | 68.8  | 64.1      |
> | READ   | 74.1  | 70.6  | 76.1  | 69.2  | 64.5      |
> | SuMi   | 74.6  | 71.2  | 76.5  | 69.6  | 64.9      |
>
> For strong OOD, we use Corrn to denote that n modalities are corrupted, miss n to denote n modalities are missing and corr+miss to denote both missing modalities and corruption modalities are present. From the table, we can observe that in dataset with more modalities, SuMi can also outperform existing method, demonstrating its effectiveness. We have added these results in our revised manuscript.

---

> ### Author Response · Authors · 2024-11-16
> **Rebuttal (2/2)**
>
> **Q3: Explanation of IQR and exploration of smoothing process.**
>
> IQR stands for interquartile range. We choose this specific quantile range because it effectively captures the central tendency and variability of the data while being less sensitive to outliers compared to other ranges. Specifically, there are three advantages:
>
> - Robustness: The IQR is less sensitive to extreme values compared to other quantile ranges, making it a more reliable measure of central tendency and variability in datasets that may contain outliers. Unlike the range, which can be affected by extreme values, the IQR focuses on the central values, making it a robust measure of variability.
>
> - Focus on Central Data: By concentrating on the middle 50% of the data, the IQR provides a clearer picture of the distribution and helps to mitigate the impact of skewed data.
>
> - Simplicity and Interpretability: The IQR is straightforward to calculate and interpret, making it an effective choice.
>
> Due to these advantages, interquartile range becomes a popular method in the statistical learning.
>
> In case of the smoothing process, we have indeed explored alternative functions for smoothing, including logarithmic and exponential functions. These alternatives are tested to assess their effectiveness in enhancing the performance of our model. In our paper, we opted for the simple linear process for its clarity. Here are the results using different smoothing functions on mixed distribution shifts of Kinetics50-C. Specifically, we use $f(t)=\log (\frac{(e-1)t}{iter}+1)$ as log function and as $f(t)=\exp{(\frac{t \ln 2}{iter})}-1$ exp function, which ensures that the value range of the function is between 0 and 1. Then the selected samples of iteration t are in [Q1-$\frac{3}{2}f(t)$IQR, Q3+$\frac{3}{2}f(t)$IQR].
>
> | f(t) | linear | log  | exp  |
> | ---- | ------ | ---- | ---- |
> | acc  | 59.1   | 58.7 | 59.5 |
>
> From the table, we can observe that using $f(t)=\exp{(\frac{t \ln 2}{iter})}-1$ function can improve the performance of the model slightly. From the properties of the exp function, it can be seen that the function grows slowly when the variable t is small and quickly when the variable is large. For log function, it grows quickly when the variable t is small and slowly when the variable is large. This indicates that slowing down the smoothing process in the initial phase helps the model's performance. Additionally, we can observe that the function will not affect the performance drastically, indicating the effectiveness of smoothing process itself. Therefore, we opted for the simple linear process in our paper for its clarity and simplicity.
>
>
>
> **Q4: Hyperparameters selection.**
>
> Thank you for your feedback. In the ablation experiments, we explore the impact of $\mu$ on the performance of the model. Although we find that adding weight to the weak modality will help to utilize the multimodal features, the performances with different $\mu$ are stable, indicating the stability of unimodal assistance strategy. For instance, the accuracy improves only around 0.1 to 0.5 when increasing $\mu$ on Kinetics50, which is small compared to adding the strategy itself. Therefore, without much hyperparameter selection, SuMi can also achieve good results. This also indicates that unimodal assistance can select high-confident samples with rich multimodal information.
>
>
>
> We have made revisions to our paper in response to your suggestions. Specific details can be found in the updated manuscript. If you have further questions, please feel free to reach out.

---

> > ### Comment · Reviewer_d57C · 2024-11-27
> >
> > Thank you to the authors for their rebuttal and for the considerable effort put into addressing our concerns. Some of my questions have been resolved. However, regarding the motivation, I find the explanation provided not entirely convincing, so I am inclined to maintain my original score. Specifically, the stated application scenario appears to be unrelated to the text modality introduced in the new experiments, and the performance improvements in video and image modalities are quite limited. A more in-depth exploration of the application scenarios for the motivation would be a valuable direction to consider.

---

> > > ### Author Response · Authors · 2024-11-28
> > > **Further clarification of application scenarios**
> > >
> > > Thank you for your response. We appreciate the opportunity to clarify application scenarios in our paper.
> > >
> > > **Q: In-depth analysis of application scenarios of our method (especially related to text modality).**
> > >
> > > In addition to the examples provided for audio and video modalities, the corrupted text modality has significant implications across various real-world applications. Here are a few scenarios where corrupted text data may arise:
> > >
> > > **Application scenarios of Social Media and Online Communications:** In environments such as social media platforms, text data can be corrupted by various factors, including typographical errors, autocorrect failures, or even the influence of emojis in place of words. For instance, users may post urgent messages that contain misspellings or shorthand, which can lead to misinterpretation.
> > >
> > > **Application scenarios of Speech Recognition Systems:** In situations where speech recognition systems convert spoken language into text, background noise, overlapping conversations, or accents can lead to inaccuracies in the transcribed text. For example, in a crowded public space, the text output from a voice command system may be corrupted due to interference from other speakers.
> > >
> > > **Application scenarios of Scanning error:** In contexts such as document scanning, the text extracted from images or PDFs may be corrupted by visual distortions, shadows, or other artifacts.
> > >
> > > Furthermore, we take our newly added experiments as an example to illustrate the importance and application of our method in detail. We use MOSI as the dataset which is a multimodal sentiment analysis dataset (audio, video and text).
> > >
> > > **Audio corruption:** Factors such as the speaker's accent, dialect, and background noise can introduce significant noise in the audio signal. For instance, in a crowded environment or during a public event, background sounds may drown out the speaker's voice, making it challenging to accurately capture the intended sentiment.
> > >
> > > **Video corruption:** The visual component can also be affected by environmental conditions. Variations in lighting (e.g., dim lighting or harsh sunlight) can alter how expressions are perceived, while the diversity of facial features among different ethnicities may lead to different interpretations of the same emotional expressions.
> > >
> > > **Text corruption:** The information extracted from the audio modality in multimodal sentiment analysis is often transcribed into text. However, when audio signals are corrupted—due to noise or other factors—this can lead to errors in speech recognition, resulting in misidentified words or phrases. Such inaccuracies in the text can manifest as "text noise," which directly influences sentiment analysis algorithms.
> > >
> > > **Strong OOD corruption:** These domain shifts often occur simultaneously across multiple modalities. For example, if background noise in the audio leads to transcription errors, the resulting text may not accurately reflect the speaker's sentiment. This scenario exemplifies strong OOD conditions, where the integrity of the data across modalities is compromised.
> > >
> > > Additionally, corruption can occur during **data transmission and preprocessing**, such as misalignment in audio-video synchronization or incorrect feature extraction of all modalities. **This type of corruption can affect any modality.**
> > >
> > > By illustrating these scenarios, we aim to underscore the prevalence of corrupted text data across different domains and our application scenarios. We believe that these situations not only enhances the motivation for our test-time adaptation methods but also emphasizes the necessity for robust models capable of handling corrupted inputs across all modalities.
> > >
> > > Regarding the performance improvements in video and image modalities, we would like to clarify that the video modality contributes less compared to other modalities in our newly added experiments on MOSI. Here are the results on MOSI.
> > >
> > > |      | Source | SuMi | No corruption |
> > > | ---- | ------ | ---- | ------------- |
> > > | Acc  | 79.8   | 80.4 | 81.2          |
> > >
> > > From the table, we can learn SuMi can bring much improvements to the model, as the accuracy is only 81.2 even without corruption. On Kinetics-50 (Table 1 in our paper) dataset, SuMi can improve the performance significantly (around +4%). Additionally, we have evaluated our method on these real-world distribution shifts in our paper in Table 7 and the results show that SuMi can significantly improve the performance (around +4%).
> > >
> > > In response to your suggestion, we have included our more detailed application scenarios and examples in Introduction of our manuscript. Thank you for your suggestion. Please feel free to reach out if you have further questions.

---

### Official Review · Reviewer_HsAx · 2024-11-02

**Soundness:** 2
**Presentation:** 1
**Contribution:** 1
**Rating:** 5
**Confidence:** 4

**Summary:**

This paper presents SuMi, a novel approach for test-time adaptation (TTA) that aims to address distribution shifts in multimodal data. Through sample selection and mutual information sharing, the authors propose an approach that adapts effectively to complex multimodal noise patterns under distribution shifts.

**Strengths:**

The paper introduces two main contributions—(1) a dual filtering mechanism to refine sample selection and (2) a mutual information loss
strategy to promote information alignment across modalities. These strategies aim to address limitations in current TTA approaches, particularly in multimodal data scenarios.
The authors validate their method across various corruption scenarios, including challenging out-of-distribution samples. This broad
testing highlights SuMi’s robustness and effectiveness under multimodal noise conditions.

**Weaknesses:**

1.Although the experiments demonstrate the method’s effectiveness, both datasets are limited to video-audio modalities. It remains unclear whether SuMi can generalize to other multimodal datasets, such as image-text pairs. This raises a question of the method’s applicability to vision-language models and other multimodal contexts (e.g., image and point cloud as in MM-TTA[1]).
2.The proposed sample selection strategy shows promising results in the ablation study. However, comparable sample selection-based approaches (e.g., DEYO, EATA) are not sufficiently compared. A comparison of DEYO/EATA with (IQR + UA) , as well as experiments
combining DEYO/EATA and MIS, would provide clearer insights into the distinct benefits of SuMi.
3.The experiments primarily focus on corruption shifts, but multimodal scenarios often encounter various other types of distribution shifts. Including datasets that reflect shifts beyond corruption could offer a more comprehensive evaluation of SuMi’s robustness.
4.The paper’s three main components—sample identification with IQR, unimodal assistance, and mutual information sharing—lack a
cohesive, unified framework. The combination of these techniques appears to address separate aspects of TTA, but the rationale for exclusively selecting these methods is insufficiently articulated. Greater clarity on the connections and unique insights of each
component could enhance the motivation and provide readers with more convincing reasoning for the proposed approach.
[1] Shin, Inkyu, et al. "Mm-tta: multi-modal test-time adaptation for 3d semantic segmentation." Proceedings of the IEEE/CVF Conference on Computer Vision and Pattern Recognition. 2022.

**Questions:**

The authors could respond to the identified weaknesses by addressing the following aspects:Limited Multimodal Data Diversity, Insufficient Comparative Analysis with Sample Selection-Based TTA Methods, Limited Scope in Addressing Diverse Distribution Shifts, Fragmented Methodological Motivation.

---

> ### Author Response · Authors · 2024-11-16
> **Rebuttal (1/2)**
>
> Thank you for your detailed and constructive comments on our paper. We appreciate your insightful comments and suggestions. Here are our responses.
>
> **Q1: Multimodal Data Diversity.**
>
> Thank you for your suggestions. To evaluate the performance of SuMi on extra datasets, we select MOSI dataset. We choose CMU-MOSI for two primary reasons. First, it includes a text modality, allowing us to work with datasets that extend beyond just video and audio. Second, MOSI encompasses three modalities—text, image, and audio—enabling us to evaluate the performance of SuMi across a dataset with more than two modalities.
>
> Here, we introduce four types of corruptions for text modality. Specifically, we introduce random deletion of word or character (RD), random insertion of word or character (RI), word shuffling (WS) and sentence permutation (SP). We have added detailed explanations for these corruptions to the appendix of our manuscript. For backbone, we use the stacked transformer blocks trained on MOSI dataset. Then, we fine-tune the model on corrupted MOSI. Here are the results.
>
> | Method | Gauss. | Shot | Impul. | Defoc. | Glass | Mot. | Zoom | Snow | Frost | Fog  | Brit. | Contr. | Elas. | Pix. | JPEG |
> | ------ | ------ | ---- | ------ | ------ | ----- | ---- | ---- | ---- | ----- | ---- | ----- | ------ | ----- | ---- | ---- |
> | Source | 79.2   | 79.1 | 79.2   | 80.1   | 79.7  | 80.4 | 79.5 | 79.4 | 79.3  | 79.8 | 80.6  | 79.6   | 80.5  | 80.5 | 80.4 |
> | EATA   | 79.4   | 79.3 | 79.2   | 80.2   | 79.9  | 80.4 | 79.6 | 79.5 | 79.3  | 79.9 | 80.6  | 79.8   | 80.6  | 80.5 | 80.6 |
> | READ   | 79.8   | 79.7 | 79.5   | 80.5   | 79.8  | 80.5 | 79.8 | 79.9 | 79.6  | 80.3 | 80.8  | 80.2   | 80.6  | 80.6 | 80.7 |
> | SuMi   | 79.9   | 80.0 | 79.8   | 80.5   | 80.4  | 80.5 | 80.0 | 80.3 | 80.1  | 80.5 | 80.7  | 80.5   | 80.7  | 80.6 | 80.8 |
>
> | Method | Gauss. | Traff. | Crowd. | Rain | Thund | Wind |
> | ------ | ------ | ------ | ------ | ---- | ----- | ---- |
> | Source | 80.5   | 80.0   | 79.9   | 80.1 | 80.0  | 80.4 |
> | EATA   | 80.5   | 80.1   | 80.2   | 80.3 | 80.1  | 80.4 |
> | READ   | 80.6   | 80.3   | 80.2   | 80.4 | 80.2  | 80.5 |
> | SuMi   | 80.8   | 80.3   | 80.3   | 80.5 | 80.4  | 80.6 |
>
> | Method | RD   | RI   | WS   | SP   |
> | ------ | ---- | ---- | ---- | ---- |
> | Source | 74.6 | 78.1 | 77.2 | 79.1 |
> | EATA   | 74.8 | 78.3 | 77.4 | 79.2 |
> | READ   | 75.1 | 78.5 | 77.8 | 79.3 |
> | SuMi   | 75.4 | 78.8 | 77.9 | 79.8 |
>
> | Method | Corr2 | Corr3 | Miss1 | Miss2 | Corr+miss |
> | ------ | ----- | ----- | ----- | ----- | --------- |
> | Source | 73.5  | 70.2  | 75.7  | 68.7  | 64.2      |
> | EATA   | 73.8  | 70.3  | 75.8  | 68.8  | 64.1      |
> | READ   | 74.1  | 70.6  | 76.1  | 69.2  | 64.5      |
> | SuMi   | 74.6  | 71.2  | 76.5  | 69.6  | 64.9      |
>
> For strong OOD, we use Corrn to denote that n modalities are corrupted, miss n to denote n modalities are missing and corr+miss to denote both missing modalities and corruption modalities are present. From the table, we can observe that in dataset with more modalities, SuMi can also outperform existing methods, demonstrating its effectiveness. We have added these results in our revised manuscript.
>
>
>
> **Q2: Insufficient Comparative Analysis with Sample Selection-Based TTA Methods.**
>
> To provide clearer insights into the distinct benefits of SuMi, we combine our strategies into other methods and present the results here.
>
> | Method | DeYo+IQR+UA | EATA+IQR+UA | DeYO+MIS | EATA+MIS |
> | ------ | ----------- | ----------- | -------- | -------- |
> | ACC    | 57.4        | 58.0        | 51.2     | 56.4     |
>
> DeYO proposes pseudo-label probability difference for sample identification which is parallel to our selection strategy (IQR+UA). EATA also proposes its own sample selection method. Therefore, we replace their sample identification methods with ours (IQR+UA). For their combinations with MIS, we just add MIS loss function to their objectives. From the table, we can observe that our strategy can enhance their performances. This indicates the effectiveness and distinct benefits of SuMi's components.

---

> ### Author Response · Authors · 2024-11-16
> **Rebuttal (2/2)**
>
> **Q3: Limited Scope in Addressing Diverse Distribution Shifts.**
>
> To evaluate the robustness of SuMi in addressing real-world distribution shift, we conduct experiments on two datasets (MOSI and CH-SIMS). Specifically, MOSI and CH-SIMS are multimodal sentiment analysis datasets which include three modalities. They contain different topics of conversations, different speakers, and different recording environments which can all be seen as real-world distribution shifts. We use stacked Transformer blocks as the backbone and pre-train the model on MOSI and CH-SIMS as the source model for the setting MOSI$\rightarrow$ CH-SIMS and CH-SIMS$\rightarrow$ MOSI, respectively. Here are the results.
>
> | Method | MOSI$\rightarrow$ CH-SIMS |      | CH-SIMS$\rightarrow$ MOSI |      |
> | ------ | ------------------------- | ---- | ------------------------- | ---- |
> |        | ACC                       | F1   | ACC                       | F1   |
> | Source | 38.7                      | 38.9 | 39.7                      | 44.8 |
> | EATA   | 40.5                      | 41.2 | 40.4                      | 45.7 |
> | READ   | 42.0                      | 42.5 | 40.9                      | 46.9 |
> | SuMi   | 44.2                      | 44.7 | 41.6                      | 47.8 |
>
> As shown in the table, we can observe that in real-world distribution shifts, SuMi can still outperform existing methods, showing its robustness.
>
> **Q4: Fragmented Methodological Motivation.**
>
> We would like to clarify that the three strategies we proposed are interconnected and not independent of each other. First, IQR is used to smooth the adaptation process to avoid abrupt huge distribution gap. IQR smoothing can help unimodal assistance identify the rich multimodal data more accurately. In the first few iterations, without IQR smoothing, it is hard for unimodal assistance to identify high-confident samples with rich information because the batch contains many strong OOD samples where the source model cannot handle well. Therefore, based on IQR smoothing, the function of unimodal assistance will be boosted, especially in mixed OOD samples. Meanwhile, IQR smoothing and unimodal assistance deal with the shift from two perspectives. IQR smoothes the adaptation process and unimodal assistance selects the data which are good for the entropy minimization. Additionally, mutual information sharing is introduced to reduce the discrepancies across different modalities and enhance the information utilization of different modalities. This is beneficial to unimodal assistance because mutual information sharing can help the unimodal branch produce more confident unimodal predictions, thus enhancing the unimodal assistance strategy. Therefore, each component is designed to complement the others, enhancing the overall effectiveness of our approach. And we can observe from the ablation experiments that the strategies combined can further improve the performance of SuMi.
>
> We have made revisions to our paper in response to your suggestions. Specific details can be found in the updated manuscript. If you have further questions, please feel free to reach out.

---

> > ### Comment · Reviewer_HsAx · 2024-11-18
> >
> > The authors' response addressed some of my concerns, but several key issues remain inadequately explained, leading me to revise my rating to 5, which is still below the acceptance threshold and insufficient for publication at ICLR.
> >
> > 1.  In Q2, I requested a comparison between Deyo/EATA and IQA+UA.  However, the experiments provided only involve adding the authors' proposed modules to Deyo/EATA, rather than the direct comparison I asked for.
> >
> > 2.  The newly added experiment with a three-modality setup shows marginal improvements.  Would tuning vision-language models, such as CLIP-type models, be more appropriate for TTA?  I believe this is more common and would better demonstrate the generalizability of your work.
> >
> > 3.  The motivation behind the paper is unclear, as the three proposed components lack strong interconnection, and the explanation provided in the rebuttal is not convincing enough.

---

> > > ### Author Response · Authors · 2024-11-20
> > > **Further Discussion (1/2)**
> > >
> > > Thank you for your response and increasing your score. We appreciate your valuable feedback and we are happy to provide further clarifications regarding our paper.
> > >
> > > **Q1: Comparison between Deyo/EATA and IQA+UA.**
> > >
> > > We apologize for any misunderstanding regarding your question. Here are the comparisons between Deyo/EATA and IQA+UA on the mixed severity level of Kinetics50-C.
> > >
> > > | Method | video-c | audio-c | strong OOD |
> > > | ------ | ------- | ------- | ---------- |
> > > | DeYo   | 71.3    | 70.4    | 22.7       |
> > > | EATA   | 71.2    | 70.1    | 38.5       |
> > > | IQR+UA | 72.4    | 72.1    | 39.6       |
> > >
> > > Our sample identification method outperforms other strategies, highlighting the effectiveness of the smoothing process and unimodal assistance. The combination of IQR smoothing and unimodal assistance allows for the selection of high-confidence samples enriched with multimodal information, effectively mitigating abrupt distribution shifts that other strategies struggle to manage in multimodal contexts. As a result, the IQR + UA approach significantly outperforms these alternative strategies.
> > >
> > >
> > >
> > > **Q2: Performance on vision-language model such as CLIP-type models.**
> > >
> > > The experiments on the MOSI dataset show only marginal improvements when the video and audio modalities are corrupted. This is primarily because, in the MOSI dataset, text is the dominant modality, and the impact of corrupted video or audio on model performance is much less significant compared to that of the text modality. Consequently, the observed improvements on MOSI are limited. However, when the text modality is corrupted, we do see noticeable enhancements in performance.
> > >
> > > To further evaluate the generalization ability of SuMi, we conduct experiments on the UPMC-Food101 dataset which is an image-text dataset for food classification. We use the CLIP image encoder and text encoder as the modality-specific encoders followed by a fusion classification head. Here are the results on Food101.
> > >
> > > | Method | Gauss. | Shot | Impul. | Defoc. | Glass | Mot. | Zoom | Snow | Frost | Fog  | Brit. | Contr. | Elas. | Pix. | JPEG |
> > > | ------ | ------ | ---- | ------ | ------ | ----- | ---- | ---- | ---- | ----- | ---- | ----- | ------ | ----- | ---- | ---- |
> > > | Source | 83.2   | 82.9 | 84.5   | 86.8   | 85.6  | 88.4 | 86.1 | 85.9 | 83.4  | 85.2 | 89.5  | 86.2   | 89.0  | 89.2 | 89.0 |
> > > | EATA   | 83.4   | 83.2 | 84.8   | 87.1   | 85.8  | 88.5 | 86.2 | 85.8 | 83.5  | 85.9 | 89.5  | 86.3   | 89.3  | 89.4 | 89.2 |
> > > | READ   | 84.2   | 83.8 | 85.6   | 87.5   | 86.3  | 88.8 | 87.3 | 86.1 | 84.1  | 86.2 | 89.4  | 86.8   | 89.2  | 89.5 | 89.5 |
> > > | SuMi   | 84.5   | 84.2 | 86.1   | 87.9   | 86.9  | 89.2 | 87.2 | 86.3 | 84.3  | 86.6 | 89.8  | 87.1   | 89.4  | 89.7 | 89.6 |
> > >
> > > | Method | RD   | RI   | WS   | SP   |
> > > | ------ | ---- | ---- | ---- | ---- |
> > > | Source | 63.8 | 81.3 | 85.1 | 87.2 |
> > > | EATA   | 63.9 | 81.9 | 85.8 | 87.4 |
> > > | READ   | 64.8 | 82.2 | 86.2 | 88.1 |
> > > | SuMi   | 67.2 | 82.5 | 86.2 | 88.2 |
> > >
> > > | Method | Both | VMiss | TMiss | Mix  |
> > > | ------ | ---- | ----- | ----- | ---- |
> > > | Source | 76.2 | 81.2  | 54.3  | 48.2 |
> > > | EATA   | 76.5 | 81.3  | 55.2  | 48.9 |
> > > | READ   | 76.7 | 81.5  | 54.8  | 48.3 |
> > > | SuMi   | 76.8 | 81.7  | 56.1  | 49.6 |
> > >
> > > From the results, we can observe that SuMi exhibits excellent generalization ability and outperforms existing methods consistently, which further demonstrates the effectiveness of SuMi.

---

> > > ### Author Response · Authors · 2024-11-20
> > > **Further Discussion (2/2)**
> > >
> > > **Q3: Further clarifications about the connection of three strategies.**
> > >
> > > Thank you for your feedback. The interplay among these strategies can be summarized as follows:
> > >
> > > - IQR smoothing establishes a controlled environment for adaptation by selecting samples that maintain stability, which is essential for the next strategies (UA) to function optimally.
> > > - Unimodal assistance operates within the framework set by IQR smoothing, ensuring that the samples chosen not only maintain stability but also contain valuable and rich multimodal information that contributes to the model's performance.
> > > - Mutual information sharing takes this a step further by ensuring that the selected samples from both IQR and unimodal strategies are aligned across modalities, thus enhancing the overall reliability of the model during adaptation.
> > >
> > > To make the explanation more convincing, we conduct several experiments. First, we present the results of the impact of MIS on the sample identification process.
> > >
> > > |          | acc using selected samples |
> > > | -------- | -------------------------- |
> > > | with MIS | 58.1                       |
> > > | w/o. MIS | 57.4                       |
> > >
> > > Please note that for the results labeled 'with MIS,' we do not incorporate MIS into the loss function; instead, we use it solely for sample selection. In other words, the only difference between the two sets of results lies in the samples used for optimization, while all other factors remain constant. We observe that MIS enhances the sample selection process and improves overall performance. This is because MIS aids the unimodal assistance strategy by enabling the unimodal branch to generate more confident predictions, which in turn strengthens the effectiveness of the unimodal assistance.
> > >
> > > Furthermore, we present the results that illustrate the relationship between IQR smoothing and unimodal assistance. Specifically, we analyze the samples selected by unimodal assistance in area 1 under three different ranges.
> > >
> > > |                 | acc using area1 samples |
> > > | --------------- | ----------------------- |
> > > | UA              | 46.2                    |
> > > | UA (within IQR) | 57.4                    |
> > > | UA (out of IQR) | 32.1                    |
> > >
> > > The results demonstrate the impact of IQR smoothing on the unimodal assistance (UA) strategy. With IQR, UA is able to select samples that not only maintain stability but also retain valuable and rich multimodal information. We believe with these experimental results, the interconnections and collective contributions of the three strategies can be more convincing. Thank you for the suggestion.
> > >
> > > We have revised our paper in response to your feedback. Thank you for your insightful comments; they have significantly enhanced the comprehensiveness and persuasiveness of our work. If you have any further questions or concerns, please feel free to reach out.

---

### Meta-Review · Area_Chair_tyNz · 2024-12-20

**Metareview:**

This paper introduces a new task, multimodal wild test-time adaptation (TTA), building upon existing wild uni-modal TTA and mild multi-modal TTA settings. The task is both interesting and practical, especially for multi-modal foundation models facing out-of-distribution (OOD) data streams. The authors conduct thorough empirical evaluations across diverse modalities (audio, video, image, text), demonstrating the applicability of the task.

The paper received mixed ratings (6, 6, 5, 5). Two of the reviewers with positive ratings expressed that their concerns had been addressed, maintaining positive scores and affirming the value of the paper. Reviewer HsAx raised concerns regarding limited multimodal data diversity, insufficient comparative analysis with sample selection-based TTA methods, a narrow scope in addressing diverse distribution shifts, and fragmented methodological motivation. In response, the authors provided a detailed rebuttal, adding new experiments with additional modalities, baseline comparisons, OOD types, and clarifications of the approach. While most of the concerns appear to have been addressed, Reviewer HsAx maintained a negative rating. Reviewer d57C, who also gave a negative rating, expressed concerns about the lack of application scenarios for the revealed task. Despite the authors’ detailed responses and clarifications, this reviewer remained unsatisfied.

Despite these two negative reviews, I agree with the two reviewers with positive ratings that the paper has undergone thorough validation and that the new task offers valuable insights for the TTA community. Therefore, I recommend acceptance.

**Additional Comments On Reviewer Discussion:**

The two reviewers with negative ratings primarily focused on the limited validations and the practical applicability of the task. In response, the authors provided detailed rebuttals, including new experimental results across additional scenarios. However, these reviewers remained dissatisfied with the revisions. In contrast, the other two reviewers felt that their concerns had been adequately addressed and maintained their positive scores.

---

### Decision · Program_Chairs · 2025-01-22

Accept (Poster)